# Spatiotemporal Backward Inconsistency Learning Gives STNNs Icing on the Cake

## Abstract

Spatiotemporal prediction models facilitate various smart-city applications across various domains,such as traffic and climate. While current advancements in these models emphasize leveraging cutting-edge technologies to enhance spatiotemporal learning, they often operate under the implicit assumption of spatiotemporal feature consistency between inputs and labels, overlooking the critical issue of historical-future inconsistency. In this study, we introduce a universal spatiotemporal backward inconsistency learning module capable of seamless integration into a variety of models, offering a notable performance boost by explicitly integrating label features to address historical-future inconsistency. Our approach includes the development of a spatiotemporal residual theory, advocating for a holistic spatiotemporal learning that encompasses both forward spatiotemporal learning to capture input data's spatiotemporal features for generating base predictions, akin to existing STNNs, and a backward process to learn residuals that rectify historical-future inconsistency, thereby refining the base predictions. Based on this theory, we design the **S**patio-**T**emporal **B**ackward **I**nconsistency Learning **M**odule (STBIM) for this backward correction process, comprising a residual learning module for decoupling inconsistency information from input representations and label representations, and a residual propagation module for smoothing residual terms to facilitate stable learning. The generated prediction correction is used to enhance the prediction accuracy. Experimental results on 11 datasets from the traffic and atmospheric domains, combined with 15 spatiotemporal prediction models, demonstrate the broad positive impact of the proposed STBIM. The code is available at https://anonymous.4open.science/r/ICLR2025-2598.

## 1 Introduction

Spatiotemporal prediction is critical for smart cities, having significant impacts in the transportation and atmospheric domains(Miao et al., 2022; Liu et al., 2024a; 2021). Current advances in spatiotemporal neural networks (STNNs) focus on crafting more expressive architectures beyond conventional models such as GCN (Kipf and Welling, 2017) or Transformer (Vaswani et al., 2017). Drawing inspiration from the fields of natural language and vision, innovative architectural concepts are being integrated into STNNs, such as the adoption of masked autoencoder (MAE) technology.

However, as the complexity of the models increases, the potential for performance gains may decrease (Shao et al., 2022a; Tang et al., 2022). In contrast to this prevailing trend, we delve into existing Spatiotemporal Neural Networks (STNNs) to uncover avenues for improvement: the majority of STNNs engage in a forward learning process to capture the spatiotemporal features of historical observations (inputs). Subsequently, the acquired representations, known as label representations, are fed into a predictor (e.g., a fully connected layer) for decoding and generating labels. This traditional approach implicitly operates under the assumption of historical-future consistency, presupposing that the spatiotemporal features of the input data and labels align seamlessly. However, this assumption is precarious, as discrepancies in spatiotemporal features between the input and labels can exist. We term such discrepancies as historical-future inconsistency. In the spatial dimension, this inconsistency can manifest in two scenarios: 1) similar input data following by different labels, and 2) different input data following by similar labels. To illustrate this concept, we depict time series data collected from two sensors (#15 and #600) in the LargeST-SD dataset (Liu et al., 2024b). In Figure 1 (a), these two sensors exhibit similar traffic flow patterns in the input data. However, in the subsequent

prediction future values, they show distinctly different patterns. Conversely, in Figure1 (b), despite differences in the distribution of traffic flow in the input data, they exhibit significant similarity in the subsequent labels. As illustrated in Figure 1 (c), inconsistencies in temporal features exist between historical and future values. Typical examples of this phenomenon include abnormal signals characterized by a rapid increase or decrease in traffic flow. STNNs encounter difficulty in accurately discerning historical-future inconsistencies, consequently leading to prediction errors. Despite the implementation of specialized techniques, such as node embedding, to support the forward learning process by some researchers (Deng et al., 2021; Shao et al., 2022a), we contend that STNNs grounded in the historical-future consistency assumption may encounter persistent challenges to effectively mitigate this inconsistency due to suboptimal modeling of label features.

We propose integrating label features into the spatiotemporal learning process to enhance the effectiveness of the model in addressing historical-future inconsistencies. Our approach involves the development of the spatiotemporal residual theory, which advocates for a bidirectional spatiotemporal learning paradigm that extends a backward process within the existing paradigm. Specifically, the theory reveals that considering label features, the final prediction should be determined by two key components: the base prediction obtained from forward spatiotemporal learning of input data features, akin to existing STNNs, termed as the base prediction, and the prediction correction term generated through learning residuals, representing historical-future inconsistencies.

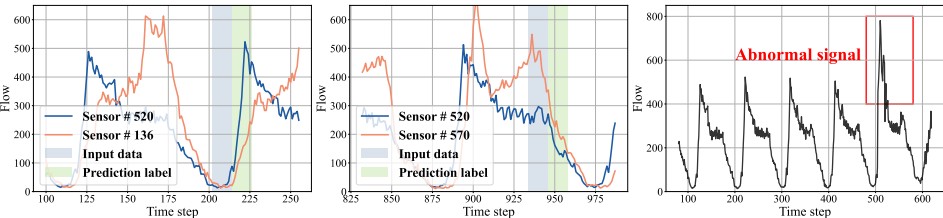

Figure 1: historical-future inconsistency in the dimensions of spatial and temporal.

Building upon this novel paradigm, we design a simple yet effective **S**patio-**T**emporal **B**ackward **I**nconsistency Learning **M**odule, namely STBIM. This module, designed to be model-agnostic, seamlessly integrates into existing STNNs to enhance performance. Specifically, any STNN is used to perform forward spatiotemporal learning, generating label representations and making base predictions. Subsequently, STBIM disentangles the residual terms by comparing the label representation with the input representation. The use of label representation enables us to model spatiotemporal inconsistency across diverse dimensions without directly accessing the labels, as they closely mirror the distribution of the labels, akin to high-dimensional feature mappings of the labels(Li et al., 2015; Shalev et al., 2018). After smoothing the generated residuals with a propagation kernel to avoid outlier signals, we decode the residuals to generate correction terms for improving accuracy by correcting base prediction. During training, STBIM can be updated end-to-end with the STNN to effectively model label-input inconsistency, while during inference, STBIM can drive STNN to generate more precise predictions. We thoroughly evaluate the effectiveness of STBIM on 12 datasets with over 11 advanced STNNs, and results demonstrate the extensive effectiveness of our module. The maximum performance increase can be up to **21.18%**.

Our contributions can be summarized as: (1) **Novel paradigm**. We develop the spatiotemporal residual theory, promoting a novel bidirectional spatiotemporal learning paradigm integrating with label features. (2) **Universal module**. STBIM, a straightforward yet potent module, seamlessly integrates with existing STNNs, which perform a backward process to explicitly model historical-future inconsistency. (3) **Thorough experiment**. We comprehensively evaluate our model on 11 commonly used spatiotemporal datasets from transportation and atmospheric domains with over 15 advanced STNNs to demonstrate the effectiveness of the STBIM module.

## 2 RELATED WORK

**Spatiotemporal prediction.** Recently, STNNs are the most representative approaches for spatiotemporal prediction tasks (Zhou et al., 2023; Wang et al., 2023; Xia et al., 2023; Huang et al., 2024; Zhou

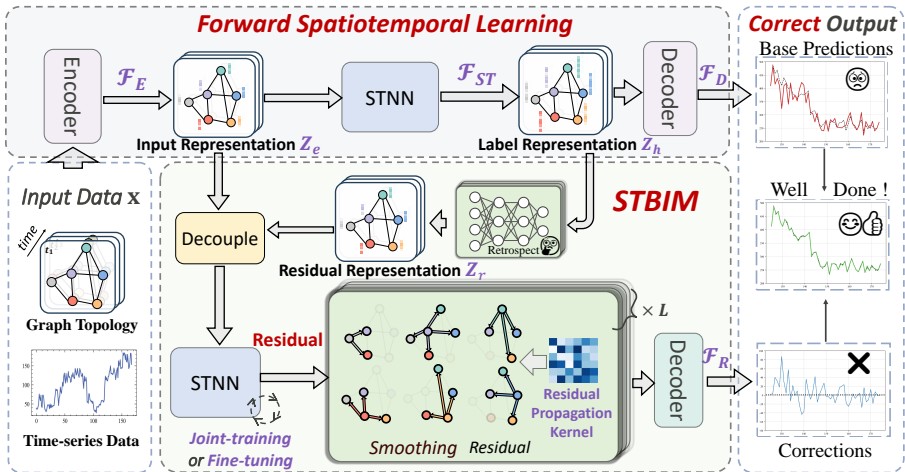

Figure 2: The overall framework of STBIM for spatiotemporal learning.

et al., 2020), which typically include a spatial module that captures spatial dependencies and a sequential module that captures temporal dependencies respectively. For example, SSTBAN (Guo et al., 2023) follows a multi-task framework by incorporating a self-supervised learner to produce robust latent representations for historical traffic data. STID (Shao et al., 2022a) identified spatiotemporal deviation phenomena and proposed utilizing node embeddings to alleviate spatiotemporal deviation. However, it did not effectively model label features, thus failing to thoroughly capture spatiotemporal inconsistencies, especially regarding temporal inconsistency features. In the experimental section, we provide a detailed comparison with STID in Appendix B.6. Some studies explore non-model approaches such as ST-LoRA (Ruan et al., 2024) to improve existing models with node-adaptive low-rank layers, the reported results show limited enhancements. Furthermore, Adaptive Graph Sparsification (AGS) (Duan et al., 2023) and Graph Winning Ticket (GWT) (Duan et al., 2024) algorithms focus on optimizing adjacency matrices in prediction models to improve operational efficiency of Adaptive Spatial-Temporal Graph Neural Networks like AGCRN. In contrast to the aforementioned work, we propose a universal module that can significantly enhance the performance of models across various tasks.

**Spatiotemporal shift learning in OOD scenario.** Traditional spatiotemporal architectures adhere to the independent and identically distributed (IID) assumption, while spatiotemporal data shift poses a challenge for out-of-distribution (OOD) generalization. Several spatiotemporal out-of-distribution (OOD) models have emerged in recent literature. For instance, CauSTG (Zhou et al., 2023) introduces a causal framework designed to transfer both local and global spatiotemporal invariant relationships to OOD scenarios. CaST (Xia et al., 2023) utilizes a structural causal model (SCM) to interpret the data generation processes of spatiotemporal graphs. Similarly, STEVE (Hu et al., 2023) encodes traffic data into two disentangled representations and incorporates spatiotemporal environments as self-supervised signals, thereby integrating contextual information into these representations. Additionally, STONE (Wang et al., 2024) proposes a causal graph structure aimed at learning robust spatiotemporal semantic graphs for OOD learning. However, while these models focus on addressing overall shifts between training and testing data, we focus on a granular shift between historical observed data (input) and predicted future data. This shift is present in both OOD and IID scenarios.

## 3 PROBLEM PRELIMINARIES

**Spatiotemporal graph data.** We use a graph $\mathcal{G} = (V, \mathcal{E}, \mathbf{A})$ to represent spatiotemporal data, where $V$ means the node set with $N$ nodes, $\mathcal{E}$ means the set of edges, and $\mathbf{A} \in \mathbb{R}^{N \times N}$ is the weighted adjacency matrix of the graph $\mathcal{G}$. We use $x_t \in \mathbb{R}^{N \times f}$ to represent the observed spatiotemporal graph data of $N$ nodes at time step $t$, where $f$ indicates the number of feature channels.

**Spatiotemporal prediction.** Given the graph $\mathcal{G}$ and the historical data of the past $T$ time steps $\mathbf{x} = \{x_1, ..., x_T\} \in \mathbb{R}^{T \times N \times f}$ as inputs, this task aims to learn a function $\mathcal{F}$ that can effectively predict the values (i.e., labels) $\mathbf{y} = \{x_{T+1}, ..., x_{T+T_P}\} \in \mathbb{R}^{T_P \times N \times f}$ in further $T_P$ time steps.

## 4 METHOD

In this section, we present details of the proposed STBIM, as shown in Figure 2 and Algorithm 1. We first develop a spatiotemporal residual theory to elucidate a comprehensive learning paradigm that considers label features. Subsequently, based on this theory, we design our model and introduce the implementation of the module. We summarize some important definitions in Table 6.

### 4.1 SPATIOTEMPORAL RESIDUAL THEORY WITH GAUSSIAN MARKOV RANDOM FIELD

The Gaussian Markov Random Field (GMRF) is a highly effective tool for modeling complex dependencies among random variables in a structured manner, which has been widely utilized in spatiotemporal dynamic analyses (Zheng and Su, 2016; Furtlehner et al., 2021). Drawing insights from these pioneering studies, we incorporate the GMRF model into our research to capture intricate relationships in spatiotemporal data. This GMRF maps spatiotemporal data points to variables and analyzes the interdependencies among these variables. Throughout the subsequent sections, we denote spatiotemporal data points using regular font and their corresponding random variables in the GMRF using *italic font*. For instance, $\mathbf{x}$ and $x$ represent spatiotemporal data and their associated variables in the GMRF, respectively.

First, we stack the input data $\mathbf{x}$ and the label $\mathbf{y}$ along the temporal dimension into a tensor $\mathbf{T} := [\mathbf{x}, \mathbf{y}] \in \mathbb{R}^{(T+T_P) \times N \times f}$. We use $\mathbf{T}_{t,:,:} \in \mathbb{R}^{N \times f}$ to denote spatiotemporal data of all nodes at $t$-th time step. We also use $\mathbf{T}_{:,u,:} \in \mathbb{R}^{(T+T_P) \times f}$ to represent the spatiotemporal data of $u$-th node during all time steps. As mentioned above, the random variable of $\mathbf{T}$ in our GMRF is denoted as $T$.

In a GMRF, all values in the matrix $\mathbf{T}$ are jointly sampled from a distribution over the random variable $T$. The joint distribution of $T$ in the GMRF is characterized by a probability density function(Baz et al., 2022; Rue and Held, 2005).

$$f_T \left( T = \mathbf{T} \mid W, \theta \right) = \frac{e^{-\Phi(\mathbf{T}|W,\theta)}}{\int d\mathbf{T}' e^{-\Phi(\mathbf{T}'|W,\theta)}}, \tag{1}$$

where $W \in \mathbb{R}^{(T+T_P) \times (T+T_P)}$ and $\theta \in \mathbb{R}^{(T+T_P)}$ are the parameters of GMRF. $W$ should be symmetric positive definite and $\theta$ is entry-wise positive. The exponent power function $\Phi$ is defined as

$$\Phi \left( \mathbf{T} \mid W, \theta \right) := \frac{1}{2} \sum_{u \in V} \mathbf{T}_{:,u,:}^\top W \mathbf{T}_{:,u,:} + \frac{1}{2} \sum_{t=1}^{T+T_P} \theta_t \mathbf{T}_{t,:,:}^\top \mathcal{A}\left(\mathbf{A}\right) \mathbf{T}_{t,:,:}, \tag{2}$$

$$= \frac{1}{2} \mathcal{V}\left(\mathbf{T}\right)^\top \mathbf{\Gamma} \mathcal{V}\left(\mathbf{T}\right), \tag{3}$$

where the potential matrix $\mathbf{\Gamma}$ reflects the dependence of variables of GRMF in temporal and spatial dimensions, which can be computed as

$$\mathbf{\Gamma} := \left(W \otimes \mathbf{I}_N\right) + \mathrm{diag}\left(\theta\right) \otimes \mathcal{A}\left(\mathbf{A}\right) \in \mathbb{R}^{[(T+T_P)N] \times [(T+T_P)N]}. \tag{4}$$

Here $\mathcal{A}\left(\mathbf{A}\right) = \mathbf{I}_N - \mathcal{N}\left(\mathbf{A}\right)$ is a graph Laplace-like operator, and $\mathbf{I_N}$ is the identity matrix of the adjacency matrix $\mathbf{A}$, $\mathcal{N}\left(\cdot\right)$ is a normalization operator such as the normalized graph Laplacian $\mathcal{A}\left(\mathbf{A}\right) = \mathbf{I_N} - \mathbf{D}^{-1/2} \mathbf{A} \mathbf{D}^{-1/2}$ with degree matrix $\mathbf{D} = \mathrm{diag}\left(\sum_{u \in V} \mathbf{A}_{1,u}, ..., \sum_{u \in V} \mathbf{A}_{N,u}\right)$ and diagonalization operator $\mathrm{diag}\left(\cdot\right)$. $\mathcal{V}\left(\cdot\right)$ is a vectorization operator to unfold the first two dimensions of the input, i.e., $\mathcal{V}\left(\mathbf{T}\right) = \left(\mathbf{T}_{1,1,:}, ..., \mathbf{T}_{T+T_P,N,:}\right)^\top \in \mathbb{R}^{[(T+T_P)N] \times f}$, and $\otimes$ is the Kronecker product.

In a Gaussian Markov Random Field (GMRF), two key parameters play important roles in modeling the relationships within the spatiotemporal graph. The parameter $\theta$ reflects the concept of homophily among nodes in the graph, indicating that a higher value of $\theta$ signifies greater compatibility among the features of nodes at the same time step. On the other hand, the parameter $W$ is responsible for controlling the level of noise present in the spatiotemporal environment. Specifically, it represents the inverse of the variance between temporal data on each node, if there is no correlation between nodes, i.e., $\mathcal{A}\left(\mathbf{A}\right) = \mathbf{0}$. For more detailed information about the parameters of the Gaussian Markov Random Field, please refer to the provided Appendix A.

**Theory 1. Forward spatiotemporal learning**. Existing spatiotemporal learning models aims to learn the conditional distribution of the future variable $y$ with respect to historical input data $\mathbf{x}$,

i.e., $\boldsymbol{y}|\boldsymbol{x} = \mathbf{x}$. In GMRF model, this goal can be be expressed in closed form as a composite of spatiotemporal operations. Specifically, for any further time step $t = \{1, 2, \cdots, T_P\}$, the expectation of the variable $\boldsymbol{y}_t \in \mathbb{R}^{1 \times N \times f}$ representing labels all nodes at $t$-th future time step with respect to the input data $\mathbf{x}$ can be computed as:

$$\mathbb{E}\left[\boldsymbol{y}_t|\mathbf{x}\right] = (1 - \gamma_t) \sum_{k=0}^{\infty} \left(\gamma_t \mathcal{N}\left(\mathbf{A}\right)\right)^k \mathbf{x}^\top \times_2 \boldsymbol{\beta}_t^\top, \tag{5}$$

$$= (1 - \gamma_t) \sum_{k=0}^{\infty} \left(\gamma_t \mathcal{N}\left(\mathbf{A}\right)\right)^k \times_2 \left(\boldsymbol{\beta}_t \mathbf{x}\right)^\top, \tag{6}$$

where $\gamma_t = \frac{\theta'_t}{W_{t',t'} + \theta'_t}$ is a scaling scalar, and $W_{t',t'}$ means the value of $t'$-th row and $t'$-th column of $W$ with $t' = T + t$. $\boldsymbol{\beta}_t = -\frac{W_{t',1:T}}{W_{t',t'}} \in \mathbb{R}^{1 \times T}$ is a coefficient vector, where $W_{t',1:T}$ means the first $T$ column and $t'$-th row of $W$. $\times_2$ implies performing tensor multiplication operations in the second dimension. Detailed proofs for this theory are provided in Appendix A.2.

Equation 5 outlines a general learning paradigm employed by existing STNNs, where features from both spatial and temporal dimensions are learned on input data $\mathbf{x}$ to generate predictions. The spatial operator $\sum_{k=0}^{\infty} \left(\gamma_t \mathcal{N}\left(\mathbf{A}\right)\right)^k$ corresponds to the graph convolution operation, utilizing Graph Convolutional Networks (GCN) (Shao et al., 2022c) or Transformers (Jiang et al., 2023) as operators. Conversely, the temporal operator $\boldsymbol{\beta}_t$ captures the relationship between the current prediction time step and the past $T$ time steps. This component typically functions as a time series model, such as those from the Temporal Convolutional Network (TCN) class (Bai et al., 2018), Recurrent Neural Network (RNN) class (Cheng et al., 2024), or Transformer class (Wang et al., 2013).

**Theory 2. Spatiotemporal residual theory**. The paradigm discussed earlier focuses solely on the spatiotemporal features of the input data, without effectively addressing the historical-future inconsistency. To tackle this issue, we are interested in exploring the integration of label features into the learning process. Let $\mathbf{y}_{t,u} \in \mathbb{R}^f$ represent the label of node $u$ at time step $t$, and denote the labels of the other nodes (excluding node $u$) as $\mathbf{y}_{t,\hat{u}} := \left[\mathbf{y}_{t,1}^\top, ..., \mathbf{y}_{t,u-1}^\top, \mathbf{y}_{t,u+1}^\top ..., \mathbf{y}_N^\top\right]^\top \in \mathbb{R}^{(N-1) \times f}$. There exist correlations between $\boldsymbol{y}_{t,u}$ and the labels of the other nodes, considering only the spatial correlation at each time step and disregarding the dependence across different time steps. Our objective is to incorporate the spatiotemporal features into the Gaussian Markov Random Field (GMRF) framework. The condition for the GMRF is to predict the variable $\boldsymbol{y}_{t,u}$ with the goal of minimizing the difference from the label $\mathbf{y}_{t,u}$. For any future time step $t = \{1, 2, \cdots, T_P\}$, the expectation of $\boldsymbol{y}_{t,u}$ with respect to $\mathbf{x}$ and $\mathbf{y}_{t,\hat{u}}$ is

$$\mathbb{E}\left[\boldsymbol{y}_{t,u}|\mathbf{x}, \mathbf{y}_{t,\hat{u}}\right] = \underbrace{\mathbb{E}\left[\boldsymbol{y}_{t,u}|\mathbf{x}\right]}_{\text{Base prediction}} + \underbrace{\underbrace{\beta_{t,u}\left(\mathbf{I}_N + \alpha_t \mathcal{A}\left(\mathbf{A}\right)\right)_{u,\hat{u}} \times_2 \underbrace{\mathbf{r}_{t,\hat{u}}}_{\text{Residual}}}_{\text{Propagation Kernel}}}_{\text{Correction}} \tag{7}$$

Detailed proofs of this theory are explained in Appendix A.2. The base prediction is generated by forward spatiotemporal learning corresponding to Theory 1. The smoothing coefficient is used to smooth the residual term for smooth learning. And $\tau_{t,u} = \left[(1 + \alpha_t)\left(1 + \alpha_t \mathcal{A}\left(\mathbf{A}\right)_{u,u}\right)\right]^{-1}$ is a scalar with $\alpha_t = \frac{\theta_{t'}}{W_{t',t'}}$ and $t' = T + t$, where $\mathcal{A}\left(\mathbf{A}\right)_{u,u}$ indicates the entry on the $u$-th row and $u$-th column of $\mathcal{A}\left(\mathbf{A}\right)$, and $\left(\mathbf{I}_N + \alpha_t \mathcal{A}\left(\mathbf{A}\right)\right)_{u,\hat{u}} \in \mathbb{R}^{1 \times (N-1)}$ is the $u$-th row of $\mathbf{I}_N + \alpha_t \mathcal{A}\left(\mathbf{A}\right)$ excluding itself. The smoothing coefficient signifies the affinity between node $u$ and the remaining nodes. In fact, it can be regarded as a part from the graph kernel. To differentiate, we call this graph kernel as residual propagation kernel. The residual term $\mathbf{r}_{t,\hat{u}}$ represents the difference between predicted expectations and labels, which is denoted as follows:

$$\mathbf{r}_{t,\hat{u}} := \mathbb{E}\left[\boldsymbol{y}_{t,\hat{u}}|\mathbf{x}\right] - \mathbb{E}\left[\mathbf{y}_{t,\hat{u}}\right] \in \mathbb{R}^{1 \times (N-1) \times f}. \tag{8}$$

In Equation 8, the base prediction expectation $\mathbb{E}\left[\boldsymbol{y}_{t,\hat{u}}|\mathbf{x}\right]$ is determined by the high-dimensional representation of the input data, while $\mathbb{E}\left[\mathbf{y}_{t,\hat{u}}\right]$ depends on the autocorrelation of the labels. Therefore, the residual term actually represents the difference in feature between the input and the label.

**Mark**. Theory 2 unveils that a holistic spatiotemporal learning paradigm integrating label information should encompass both a forward process and a backward process. The forward process in spatiotemporal learning captures the interdependencies in the input data to produce base predictions, whereas the backward process is dedicated to modeling residuals for generating correction terms. These residuals encapsulate the discrepant features of the input and labels. The ultimate prediction is derived from the amalgamation of the base prediction and the correction terms. This theory aligns perfectly with our assertion.

## 4.2 STBIM

We initially outline the general structure of conventional STNNs utilized for forward spatiotemporal learning. Subsequently, we provide a comprehensive description of our innovative STBIM module and elucidate its seamless integration with STNNs.

### 4.2.1 FORWARD SPATIOTEMPORAL LEARNING

As shown in Figure 2, existing spatiotemporal prediction models typically consist of three parts: (1) An input encoder maps the input data into a high-dimensional feature space, generally combining enhancement strategies such as node embedding technology. This function is denoted as $\mathcal{F}_E : \mathbf{x} \mapsto \mathbf{Z}_e \in \mathbb{R}^{T \times N \times d_e}$, where $\mathbf{Z}_e$ is termed as the *input representation*; (2) A STNN module $\mathcal{F}_{ST}$ is used to capture spatiotemporal features of input representation and generate the *label representation* $\mathbf{Z}_h$: $\mathcal{F}_{ST} : \mathbf{Z}_e \mapsto \mathbf{Z}_h \in \mathbb{R}^{T \times N \times d_h}$. The trained label representations $\mathbf{Z}_h$ can approximate the high-dimensional feature mapping of the labels; (3) A base decoder $\mathcal{F}_D$ decodes the label representation $\mathbf{Z}_h$ to generate base prediction $\mathbf{y}_{base}$, $\mathcal{F}_D : \mathbf{Z}_h \mapsto \mathbf{y}_{base} \in \mathbb{R}^{T_P \times N \times f}$.

### 4.2.2 BACKWARD SPATIOTEMPORAL INCONSISTENCY LEARNING

**Residual learning.** In Equation 8, the residual term delineates the feature disparity between the input and the labels. Our methodology involves modeling the input representation $\mathbf{Z}_e$ and the label representation $\mathbf{Z}_h$ to calculate this residual. The initial label representations are derived through spatiotemporal learning of input data. Post-training, these representations closely mirror the distribution of the labels, akin to high-dimensional feature mappings of the labels. Leveraging these representations for residual computation allows us to capture inconsistencies across diverse dimensions. Furthermore, this strategy alleviates the requirement to access the real labels, particularly in situations where acquiring the true labels is impractical during the inference phase.

Specifically, we employ a Multi-Layer Perceptron (MLP) with Gaussian Error Linear Units (GELU) activation function (Hendrycks and Gimpel, 2016; Devlin et al., 2018) to map the label representation $\mathbf{Z}_h$ to the same space: $\mathbf{Z}_h \mapsto \mathbf{Z}_r \in \mathbb{R}^{T \times N \times d_e}$. Subsequently, we decouple the spatiotemporal inconsistent features by subtracting the two representations $\mathbf{Z}_e - \mathbf{Z}_r$, which helps filter out redundant spatiotemporal features. This deviation is then fed into the STNN module $\mathcal{F}_{ST}$, placing particular emphasis on the model's relearning of this information. Consequently, the resulting output $\mathbf{Z}_{res}$ represents the residual term. The overall calculation process can be outlined as follows:

$$\mathbf{Z}_{res} = \mathcal{F}_{ST} \left( \mathbf{Z}_e - \text{MLP} \left( \mathbf{Z}_h \right) \right). \tag{9}$$

**Residual propagation kernel.** As show in Theory 2, it is essential to use spatiotemporal correlation to smooth this residual term $\mathbf{Z}_{res}$. The smoothing process is similar to an aggregation process based on a residual propagation kernel, i.e., graph kernel. In this paper, we thoroughly investigate the effectiveness of different types of graph kernels, including predefined kernel, diffusion kernel, adaptive kernel, and data-driven kernel; and their definition is obtained in Appendix A.3. In order to enhance the representation ability of models, we deploy $K$ kernels $(\mathcal{K}_1, \mathcal{K}_2, ..., \mathcal{K}_K)$. Let $\mathcal{K} := \boldsymbol{\tau} \left( \mathbf{I}_N + \frac{1}{K} \sum_{i=1}^{K} \boldsymbol{\alpha_i} \mathcal{K}_i \right)$, where $\boldsymbol{\alpha}_i = \text{diag} \left( \alpha_{i,1}, \alpha_{i,2}, ..., \alpha_{i,N} \right) \in (-1, 1)^{N \times N}$ and $\boldsymbol{\tau} = \text{diag} \left( \tau_1, \tau_2, ..., \tau_N \right) \in (0, 1)^{N \times N}$ are the learnable parameters. The variable $a$ represents the intensity of residual diffusion across the global spatiotemporal graph originating from a particular node, indicated by its magnitude. The sign of $a$ determines whether the correlation between nodes is positive or negative, thereby mitigating the risk of weak regression caused by an overly strong assumption of homogeneity for anomalous data nodes (Xu et al., 2021; Kim et al., 2022). The parameter $\boldsymbol{\tau}$ plays a crucial role in determining the overall magnitude of the residual impact.

In fact, this residual propagation under this combination of kernels is equivalent to the *linear Graph Convolution* (Wu et al., 2019a; He et al., 2020a) operation with residual skip connections. We denote the residual propagation layer as $\mathcal{F}_{RP}$, and if we denote the number of residual propagation layers as $L$, we get the smoothed residual $\tilde{\mathbf{Z}}_{res}$ along the overall spatiotemporal structure as

$$\tilde{\mathbf{Z}}_{res} = \left[ \boldsymbol{\tau} \left( \mathbf{I}_N + \frac{1}{K} \sum_{i=1}^{K} \boldsymbol{\alpha}_i \mathcal{K}_i \right) \times_2 \mathbf{Z}_{res} \right]^{L}. \tag{10}$$

**Prediction correction.** We employ a MLP layer with the GELU activation function as a residual decoder $\mathcal{F}_R$ to generate a prediction correction term $\mathbf{y}_{cor} = \mathcal{F}_R \left( \tilde{\mathbf{Z}}_{res} \right)$. Finally, this correction is added into the base prediction $\mathbf{y}_{base}$ to yield final prediction $\hat{\mathbf{y}}$:

$$\hat{\mathbf{y}} = \mathbf{y}_{base} + \mathbf{y}_{cor}. \tag{11}$$

**Training strategy of STBIM.** We introduce two training strategies for STBIM: "Joint Training" and "Fine-tuning." In Joint Training, STBIM and STNN are trained end-to-end from scratch. The fine-tuning method involves fine-tuning both pre-trained STNN and STBIM together.

## 5 EXPERIMENT

In this section, we evaluate the effectiveness of the proposed generic modules across eleven datasets and over fifteen baselines. We primarily address the following potential concerns: **Q.1** Does the proposed module enhance performance prediction for existing STNNs? **Q.2** How sensitive is the model to hyperparameters? **Q.3** Can the modules effectively handle historical-future inconsistency? Furthermore, in the appendix, we detail the comparison between STID and STBIM, the computational costs of STBIM, and its additional convergence computation benefits.

**Datasets.** We deploy experiments on 11 datasets from two domains: transportation and atmosphere. In the transportation domain, we cover several commonly used PeMS0X (X=03, 04, 07, 08), PEMS3-Stream (Chen et al., 2021), and METR-LA datasets, as well as an emerging dataset called LargeST (Liu et al., 2023b), which consists of three sub-datasets, including an extremely large-scale dataset with 8,600 nodes. The KnowAir (Wang et al., 2020) dataset records 4-year PM2.5 features from 184 atmospheric monitoring stations, and we further spilt KnowAir into 3 sub-datasets. The details of these datasets are shown in Table 1. We divide traffic datasets into training, validation, and test sets along time dimension with a ratio of 6:2:2. More details of training/validation/test sets can be found in Appendix B.1.

**Implementation.** We use the AdamW optimizer (Loshchilov and Hutter, 2017) with a learning rate of 0.002 for optimizing. To assess the efficacy of our framework, we employ commonly utilized Mean Absolute Error (MAE), Root Mean Square Error (RMSE), and Mean Absolute Percentage Error (MAPE) as metrics. The models are executed on a Nvidia A100 with 40GB memory, and the code environment is based on the PyTorch framework using Python 3.8.3. The length of the input time window and future prediction window are both set to 12 in traffic datasets and 24 in atmosphere datasets. *When training STNNs with STBIM, we maintain the hyperparameters of STNNs, ensuring that the performance gain comes only from STBIM.*

Table 1: Summary of spatiotemporal datasets.

| Dataset | Nodes | Edges | Frames |
|---|---|---|---|
| LargeST-SD | 716 | 17,319 | 525,888 |
| LargeST-GBA | 2,352 | 61,246 | 525,888 |
| LargeST-GLA | 3,834 | 98,703 | 525,888 |
| LargeST-CA | 8,600 | 201,363 | 525,888 |
| PEMS03 | 358 | 546 | 26,208 |
| PEMS04 | 307 | 338 | 16,992 |
| PEMS07 | 883 | 865 | 28,224 |
| PEMS08 | 170 | 276 | 17,856 |
| METR-LA | 207 | 1,515 | 34,272 |
| PEMS3-Stream | 655 | 1,577 | 8,928 |
| KnowAir | 184 | 3,796 | 3,4380 |

**Baselines.** We eployed STBIM into a dozen STNNs to evaluate the efficacy. These baselines consist of various models including LSTM, STGCN (Yu et al., 2017), STNN (He et al., 2020b), ASTGCN (Guo et al., 2019), STAEFormer (Liu et al., 2023a), AGCRN (Bai et al., 2020), STID (Shao et al., 2022b), GC-LSTM (Qi et al., 2019), PM2.5GNN (Wang et al., 2020), nodesFC-GRU (Wang et al., 2020), stemGNN (Cao et al., 2020), STWA (Cirstea et al., 2022), D$^2$STGNN (Shao et al., 2022d), DGCRN (Li et al., 2023), DDGCRN (Weng et al., 2023), and BigST (Han et al., 2024). All models are executed using the hyperparameters outlined in the official code (Liu et al., 2023b; Wang et al., 2020). Further information regarding these baselines can be found in Appendix B.2.

Table 2: Prediction performance of models on traffic datasets. We sequentially report the performance of each model without STBIM modules, with STBIM modules in joint-training manner (+JT), and with STBIM modules in fine-tuning manner (+FT). 'Average improvement' reports the improvement of average prediction performance during 12 time steps using STBIM relative to basemodels.

| | | LargeST-SD dataset | | | | | | | | | | | |
| --- | --- | --- | --- | --- | --- | --- | --- | --- | --- | --- | --- | --- | --- |
| Method | STBIM | Horizon 3 | | | Horizon 6 | | | Horizon 12 | | | Average improvement | | |
| | | MAE | RMSE | MAPE | MAE | RMSE | MAPE | MAE | RMSE | MAPE | MAE | RMSE | MAPE |
| LSTM | - | 19.13 | 30.80 | 11.62 | 26.07 | 41.34 | 16.32 | 37.87 | 59.37 | 25.08 | - | - | - |
| | +JT | 17.55 | 27.97 | 11.42 | 21.71 | 34.64 | 14.68 | 26.91 | 44.06 | 19.44 | +19.50% | +18.20% | +13.37% |
| | +FT | 18.68 | 30.07 | 11.52 | 24.79 | 39.08 | 15.94 | 35.20 | 54.96 | 23.86 | +5.33% | +5.73% | +3.24% |
| STID | - | 15.39 | 25.71 | 9.90 | 18.05 | 30.53 | 12.02 | 22.01 | 39.06 | 15.35 | - | - | - |
| | +JT | 14.66 | 24.71 | 9.45 | 17.08 | 29.03 | 11.20 | 20.83 | 35.64 | 14.31 | +5.11% | +5.95% | +6.06% |
| | +FT | 15.15 | 25.35 | 9.84 | 17.00 | 29.99 | 11.89 | 21.61 | 38.15 | 15.01 | +1.61% | +1.92% | +1.24% |
| STAEFormer | - | 15.68 | 25.71 | 10.65 | 18.33 | 30.42 | 12.66 | 22.77 | 38.64 | 15.99 | - | - | - |
| | +JT | 15.48 | 25.92 | 10.14 | 17.91 | 30.31 | 11.86 | 21.61 | 37.13 | 14.96 | +3.09% | +1.40% | +4.89% |
| | +FT | 15.46 | 25.66 | 10.31 | 18.14 | 30.42 | 12.18 | 21.90 | 38.17 | 15.11 | +2.22% | +0.68% | +3.63% |
| STGCN | - | 17.37 | 29.91 | 12.36 | 19.29 | 33.36 | 13.39 | 22.99 | 40.28 | 15.80 | - | - | - |
| | +JT | 16.05 | 27.39 | 10.80 | 18.30 | 31.45 | 12.21 | 22.17 | 39.22 | 15.12 | +5.73% | +5.77% | +8.79% |
| | +FT | 15.97 | 27.52 | 10.76 | 18.25 | 31.67 | 12.34 | 22.07 | 39.27 | 15.36 | +5.94% | +5.24% | +8.13% |
| STTN | - | 18.11 | 28.92 | 11.33 | 21.26 | 34.33 | 13.30 | 25.78 | 41.43 | 17.06 | - | - | - |
| | +JT | 15.94 | 25.76 | 10.66 | 18.49 | 30.26 | 12.33 | 22.54 | 38.58 | 16.10 | +12.18% | +9.85% | +6.12% |
| | +FT | 16.16 | 26.50 | 10.60 | 18.62 | 31.10 | 12.52 | 22.65 | 39.29 | 15.62 | +11.27% | +7.30% | +6.49% |
| ASTGCN | - | 20.23 | 32.17 | 13.09 | 25.94 | 40.54 | 17.13 | 32.34 | 50.86 | 22.23 | - | - | - |
| | +JT | 17.41 | 28.24 | 11.17 | 20.92 | 34.05 | 13.59 | 24.80 | 40.51 | 17.22 | +18.74% | +16.00% | +21.18% |
| | +FT | 18.58 | 29.84 | 12.31 | 23.43 | 37.90 | 15.96 | 29.70 | 48.73 | 21.58 | +8.36% | +5.44% | +6.92% |
| AGCRN | - | 15.57 | 28.49 | 11.39 | 17.66 | 31.44 | 12.86 | 21.40 | 40.44 | 16.35 | - | - | - |
| | +JT | 15.14 | 25.49 | 10.16 | 17.39 | 29.70 | 11.70 | 20.95 | 29.96 | 14.91 | +2.03% | +7.39% | +9.57% |
| | +FT | 15.52 | 28.51 | 11.48 | 17.65 | 31.38 | 12.79 | 21.39 | 40.42 | 16.40 | +0.17% | +0.15% | +0.08% |
| DGCRN | - | 15.83 | 28.48 | 12.60 | 20.50 | 33.24 | 14.08 | 24.16 | 40.67 | 16.68 | - | - | - |
| | +JT | 15.28 | 25.45 | 10.98 | 17.33 | 29.27 | 11.55 | 21.20 | 36.57 | 14.55 | +1.14% | -0.07% | +4.30% |
| | +FT | 15.72 | 25.39 | 10.28 | 17.41 | 29.20 | 11.01 | 21.10 | 36.23 | 14.19 | +1.48% | +1.65% | +0.18% |
| DDGCRN | - | 15.64 | 29.23 | 11.12 | 18.34 | 33.19 | 12.82 | 22.79 | 40.97 | 16.46 | - | - | - |
| | +JT | 15.59 | 28.00 | 11.10 | 18.13 | 31.53 | 11.99 | 22.10 | 39.17 | 15.02 | +1.73% | +3.80% | +5.95% |
| | +FT | 15.66 | 29.35 | 11.11 | 18.34 | 33.26 | 12.77 | 22.63 | 40.78 | 16.09 | +0.06% | -0.09% | +0.23% |
| D²STGNN | - | 14.93 | 25.29 | 10.37 | 17.40 | 29.69 | 12.16 | 21.31 | 36.30 | 14.99 | - | - | - |
| | +JT | 14.83 | 24.68 | 9.79 | 17.23 | 28.72 | 11.17 | 20.61 | 34.42 | 13.58 | +1.41% | +3.55% | +8.11% |
| | +FT | 14.89 | 24.83 | 9.74 | 17.42 | 28.81 | 11.31 | 20.81 | 34.58 | 13.81 | +1.21% | +3.56% | +6.02% |
| BigST | - | 15.83 | 26.04 | 11.38 | 18.17 | 31.13 | 13.12 | 22.92 | 39.63 | 16.34 | - | - | - |
| | + JT | 14.39 | 24.27 | 10.24 | 17.56 | 29.09 | 11.78 | 21.01 | 36.01 | 14.88 | +8.63% | +6.52% | +10.31% |
| | + FT | 14.62 | 24.41 | 10.50 | 17.90 | 29.48 | 12.16 | 21.17 | 36.20 | 14.94 | +7.08% | +5.913% | +7.92 |

| | | LargeST-GBA dataset | | | | | | | | | | | |
| --- | --- | --- | --- | --- | --- | --- | --- | --- | --- | --- | --- | --- | --- |
| Method | STBIM | Horizon 3 | | | Horizon 6 | | | Horizon 12 | | | Average improvement | | |
| | | MAE | RMSE | MAPE | MAE | RMSE | MAPE | MAE | RMSE | MAPE | MAE | RMSE | MAPE |
| LSTM | - | 20.21 | 33.22 | 15.14 | 27.28 | 43.34 | 23.08 | 38.55 | 60.13 | 36.68 | - | - | - |
| | +FT | 17.67 | 31.24 | 15.14 | 24.47 | 38.37 | 22.09 | 31.77 | 49.32 | 32.92 | +11.89% | +12.70% | +6.34% |
| | +JT | 19.34 | 31.70 | 14.96 | 24.71 | 39.26 | 21.90 | 32.18 | 50.35 | 33.26 | +11.06% | +10.92% | +6.34% |
| STID | - | 17.80 | 29.56 | 14.32 | 21.04 | 34.76 | 17.28 | 25.23 | 42.22 | 21.48 | - | - | - |
| | +JT | 17.43 | 29.35 | 13.37 | 20.43 | 34.19 | 15.94 | 24.35 | 40.90 | 19.92 | +2.74% | +1.88% | +7.08% |
| | +FT | 17.67 | 29.65 | 13.71 | 20.77 | 34.80 | 16.51 | 24.79 | 41.75 | 20.60 | +1.25% | +0.43% | +4.15% |
| STAEFormer | - | 18.55 | 29.94 | 14.99 | 21.69 | 34.65 | 16.87 | 26.42 | 41.50 | 21.31 | - | - | - |
| | +JT | 18.05 | 29.40 | 14.43 | 21.18 | 34.04 | 16.17 | 25.76 | 40.80 | 20.80 | +3.46% | +1.16% | +1.93% |
| | +FT | 18.32 | 29.84 | 14.15 | 21.25 | 34.15 | 16.91 | 25.66 | 40.83 | 21.34 | +1.94% | +0.96% | +1.05% |
| STGCN | - | 20.47 | 33.85 | 15.26 | 22.75 | 37.49 | 17.03 | 25.51 | 42.13 | 19.80 | - | - | - |
| | +JT | 19.38 | 32.14 | 14.52 | 22.16 | 36.65 | 16.73 | 25.61 | 42.62 | 20.04 | +2.31% | +2.02% | +3.25% |
| | +FT | 20.22 | 33.55 | 15.12 | 22.56 | 37.40 | 16.91 | 25.44 | 42.12 | 19.79 | +0.80% | +0.40% | +0.53% |
| STTN | - | 18.92 | 30.48 | 15.25 | 22.31 | 35.50 | 18.88 | 26.59 | 42.58 | 23.35 | - | - | - |
| | +JT | 18.70 | 29.98 | 15.07 | 21.99 | 35.02 | 17.89 | 26.05 | 42.06 | 22.49 | +1.73% | +1.45% | +4.27% |
| | +FT | 18.81 | 29.37 | 14.98 | 21.34 | 35.01 | 17.16 | 26.03 | 42.04 | 22.70 | +2.14% | +2.67% | +3.89% |
| ASTGCN | - | 21.53 | 34.07 | 17.44 | 26.31 | 40.36 | 24.71 | 34.00 | 52.97 | 30.15 | - | - | - |
| | +JT | 19.91 | 31.60 | 16.29 | 24.73 | 38.52 | 21.18 | 30.51 | 47.44 | 27.73 | +7.01% | +5.35% | +9.04% |
| | +FT | 20.44 | 33.63 | 16.52 | 25.14 | 39.77 | 24.66 | 32.96 | 50.67 | 29.80 | +0.61% | +1.00% | +2.76% |
| AGCRN | - | 18.04 | 30.11 | 13.99 | 20.79 | 34.29 | 16.33 | 24.28 | 39.65 | 20.21 | - | - | - |
| | +JT | 16.80 | 28.56 | 12.52 | 20.10 | 33.45 | 14.98 | 23.73 | 39.10 | 18.79 | +2.00% | +1.80% | +6.95% |
| | +FT | 17.90 | 29.99 | 12.99 | 20.75 | 34.27 | 15.41 | 24.24 | 39.01 | 19.17 | +0.29% | +0.38% | +5.91% |
| DGCRN | - | 18.02 | 28.97 | 15.23 | 21.09 | 33.88 | 18.13 | 25.86 | 41.02 | 23.66 | - | - | - |
| | +JT | 17.86 | 28.43 | 15.10 | 20.89 | 32.76 | 18.03 | 25.69 | 40.89 | 22.35 | +1.38% | +0.57% | +4.67% |
| | +FT | 17.89 | 28.54 | 15.13 | 20.77 | 32.55 | 17.98 | 25.66 | 40.76 | 22.56 | +1.52% | +0.45% | +4.23% |
| DDGCRN | - | 17.86 | 29.11 | 15.26 | 21.05 | 33.86 | 18.07 | 25.62 | 41.15 | 23.55 | - | - | - |
| | +JT | 17.47 | 28.22 | 15.24 | 20.72 | 32.74 | 18.00 | 25.57 | 40.81 | 22.19 | +2.64% | +0.81% | +5.49% |
| | +FT | 17.67 | 28.60 | 14.99 | 20.81 | 32.50 | 17.93 | 25.42 | 40.59 | 23.55 | +1.61% | +0.75% | +0.11% |
| D²STGNN | - | 17.23 | 29.91 | 12.22 | 20.50 | 34.79 | 14.96 | 25.13 | 41.8 | 19.67 | - | - | - |
| | +JT | 16.83 | 29.27 | 11.97 | 20.32 | 34.35 | 14.56 | 24.81 | 41.21 | 19.27 | +1.03% | +1.19% | +5.13% |
| | +FT | 17.06 | 29.59 | 12.01 | 20.37 | 34.64 | 14.71 | 24.92 | 41.72 | 19.25 | +0.31% | +1.26% | +1.72% |
| BigST | - | 18.29 | 29.79 | 15.18 | 21.98 | 35.39 | 18.91 | 26.74 | 43.48 | 23.91 | - | - | - |
| | + JT | 17.37 | 29.26 | 13.78 | 20.88 | 33.87 | 15.89 | 24.79 | 41.52 | 20.32 | +17.36% | +3.48% | +15.50% |
| | + FT | 17.35 | 29.44 | 13.74 | 20.85 | 33.36 | 15.93 | 24.91 | 40.94 | 20.28 | +17.44% | +4.39% | +15.01% |

## 5.1 ANALYSIS OF EXPERIMENT RESULTS (Q.1)

We report the performance of the proposed STBIM combined with various STNNs on LargeST-SD and -GBA datasets in Table 2. Note that due to space constraints, experiment analysis on the other datasets can be obtained in Appendix B.5.

The AGCRN model demonstrates low prediction errors due to its adaptive graph learning strategy, enabling the model to accurately capture spatial dependencies. Interestingly, the MLP-based architecture STID achieves competitive predictions, possibly attributed to its utilization of spatial identity encoding, which enhances the representation of spatial node embeddings. When the spatiotemporal prediction baselines are combined with the proposed STBIM module, all models exhibit improved performance. Particularly noteworthy is the significant performance enhancement observed in certain underperforming models, with ASTGCN showing an average improvement of 15% on the LargeST-SD dataset with the integration of STBIM. Even for competitive baselines like STID and AGCRN, our module yields substantial performance gains. Notably, even for complex models such as $D^2$STGNN and DDGCRN, the benefits of our module remain effective.

By comparing the training strategies of two STBIMs, it is evident that in various scenarios, the joint training approach generally outperforms the alternative methods. This advantage can be attributed to the stronger adaptability provided by joint training. In conclusion, experimental results demonstrate that our module significantly enhances the prediction accuracy of spatiotemporal prediction models in a wide range of scenarios. This improvement stems from modeling the inconsistent features between input and labels, rather than simply increasing the parameter size (as analyzed in Appendix C.0.1).

## 5.2 HYPERPARAMETER SENSITIVITY ANALYSIS (Q.2)

**The number of residual propagation layers $L$.** We evaluate the sensitivity of $L$ in Equation 10. Taking STID and STGCN as examples, experiment results on the LargeST-SD dataset are reported in Figure 3. We can see that optimal values of these two models are 2 and 3, respectively. When $L$ is smaller than the optimal value, shallow residual propagation may not effectively propagate sufficient spatiotemporal information of labels. If $L$ is equal to 0, it means that we do not utilize label information, and the large prediction errors also prove the validity of STBIM. On the other hand, when $L$ exceeds the optimal value, excessive smoothing of information may occur due to deep layers. Particularly, when $L$ is excessively large, STBIM may have a negative impact, potentially due to overfitting caused by increased model complexity.

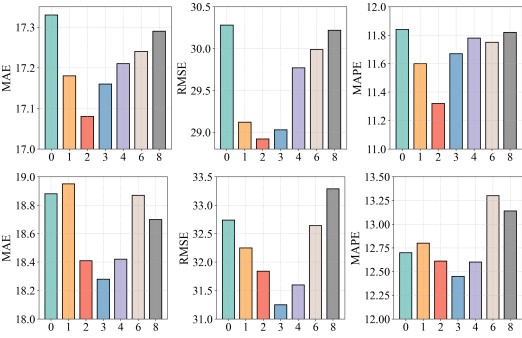

Figure 3: Hyperparameter experiment of $L$ with STID (Upper) and STGCN (lower).

Table 3: Kernel function sensitivity analysis.

| Kernel | STID | | |
|---|---|---|---|
| | MAE | RMSE | MAPE |
| Transition | 17.63 | 30.40 | 11.76 |
| DoubleTransition | 17.40 | 29.56 | 11.44 |
| Adaptive | **17.12** | **28.59** | **11.19** |
| Data-driven | 17.99 | 30.14 | 12.84 |

| Kernel | STGCN | | |
|---|---|---|---|
| | MAE | RMSE | MAPE |
| Transition | 19.04 | 33.53 | 13.42 |
| DoubleTransition | 18.85 | 32.99 | 12.90 |
| Adaptive | **18.56** | **32.34** | **12.68** |
| Data-driven | 19.30 | 32.94 | 13.17 |

**Kernel function.** We evaluate the effect of different graph kernel types on model performance, which is explained in Equation 3. The definitions of these kernels are described in Appendix A.3. We take STGCN and STID as examples, and the results are shown in Table 3. We find that adaptive kernel function for residual propagation achieves more accurate performance for both models. The underlying reason is that it can capture a more comprehensive spatiotemporal information.

## 5.3 EFFECTIVENESS ANALYSIS FOR HISTORICAL-FUTURE INCONSISTENCY (Q.3)

We assess STBIM's effectiveness in addressing inconsistencies in spatial and temporal dimensions. Temporal inconsistency is evaluated based on samples where the increase ratio of input data mean compared to label mean exceeds 75%. Spatial inconsistency is identified when the similarity of input sequences between two nodes ranks in the top 20%, while their predicted label similarity falls within the lowest 5%. The experimental results for the Large-SD datasets, as presented in Tables 4 and 5, indicate that while some models attempt to enhance node uniqueness representation through node embedding, existing high-level architectures still struggle to effectively manage non-consistent samples due to the input-label consistency assumption. Our models improve label features by explicitly modeling them. The prediction visualizations are illustrated in Figure 4. For a more detailed comparison between STID and STBIM, please refer to Appendix B.6.

Table 4: Temporal inconsistency modeling.

| Model | MAE | RMSE | MAPE |
|---|---|---|---|
| STGCN | 27.67 | 40.42 | 45.20 |
| +STBIM | **24.34** | **37.60** | **42.27** |
| STID | 27.06 | 41.40 | 43.63 |
| +STBIM | **20.55** | **32.16** | **33.50** |
| STAEformer | 25.63 | 36.06 | 35.26 |
| +STBIM | **21.79** | **34.19** | **33.91** |
| D$^2$STGNN | 21.09 | 33.37 | 34.64 |
| +STBIM | **20.31** | **31.37** | **32.85** |

Table 5: Spatial inconsistency modeling.

| Model | MAE | RMSE | MAPE |
|---|---|---|---|
| STGCN | 29.43 | 43.61 | 46.34 |
| +STBIM | **25.12** | **39.89** | **43.10** |
| STID | 25.78 | 39.15 | 28.68 |
| +STBIM | **23.06** | **38.22** | **24.31** |
| STAEformer | 27.71 | 35.12 | 26.40 |
| +STBIM | **23.89** | **34.19** | **22.91** |
| D$^2$STGNN | 24.31 | 34.62 | 24.71 |
| +STBIM | **21.09** | **33.74** | **21.13** |

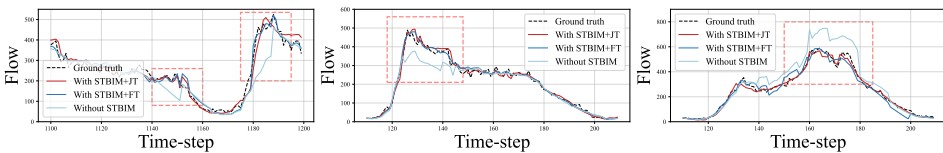

(a) Visualization cases of historical-future inconsistency in the temporal dimension

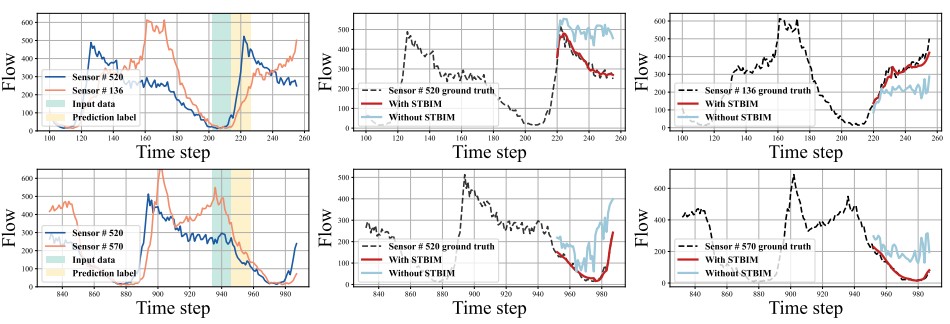

(b) Visualization cases of historical-future inconsistency in the spatial dimension

Figure 4: Visualization cases of historical-future inconsistency.

## 6 CONCLUSION

In this research, we introduce a versatile module named STBIM designed to boost the predictive capabilities of STNNs. STBIM effectively integrates label information into spatiotemporal learning by utilizing residuals. Initially, it separates the residual elements from the input and labels. It then refines these residuals by incorporating spatiotemporal correlations. Finally, the module leverages the enhanced residuals to adjust the predictions, thereby improving the model's accuracy. By integrating the STBIM module into various spatiotemporal prediction models and conducting comprehensive experiments, we observed substantial performance enhancements of up to 21.18%.

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

# A  MATHEMATICAL PROPOSITIONS AND PROOFS

## A.1  SPATIOTEMPORAL MULTIVARIATE GAUSSIAN DISTRIBUTION

**Multivariate Gaussian distribution.** Consider the multivariate Gaussian distribution (Goodman, 1963) over spatiotemporal variables $\boldsymbol{T} \sim \mathcal{N}(\bar{\mathbf{T}}, \boldsymbol{\Sigma})$, including historical and future temporal variables, where $\bar{\mathbf{T}}$ is the expectation and $\boldsymbol{\Sigma}$ is the covariance matrix. The probability density of $\boldsymbol{T}$ is

$$f_{\boldsymbol{T}}\left(\mathbf{T}|\bar{\mathbf{T}}, \boldsymbol{\Sigma}\right) = \frac{\det\left(\boldsymbol{\Sigma}\right)^{\frac{1}{2}} \exp\left(-\frac{1}{2}\left(\mathcal{V}\left(\mathbf{T}\right) - \mathcal{V}\left(\bar{\mathbf{T}}\right)\right)^{\top} \boldsymbol{\Sigma}^{-1}\left(\mathcal{V}\left(\mathbf{T}\right) - \mathcal{V}\left(\bar{\mathbf{T}}\right)\right)\right)}{\left(\sqrt{2\pi}\right)^{(T+T_P)N}}. \tag{12}$$

In the statements and proofs in Appendix, we use the bold subscript symbols $\boldsymbol{x}$ and $\boldsymbol{y}$ to denote the subtensor of the tensor corresponding to the rows and columns of the $\boldsymbol{x}$ and $\boldsymbol{y}$ parts of $\boldsymbol{T}$. Split the variable into historical and future temporal part, then the spatiotemporal variables has the distribution in the block form,

$$[\boldsymbol{x}, \boldsymbol{y}] \sim \mathcal{N}\left([\bar{\mathbf{x}}, \bar{\mathbf{y}}], \begin{bmatrix} \boldsymbol{\Sigma}_{\boldsymbol{xx}} & \boldsymbol{\Sigma}_{\boldsymbol{xy}} \\ \boldsymbol{\Sigma}_{\boldsymbol{yx}} & \boldsymbol{\Sigma}_{\boldsymbol{yy}} \end{bmatrix}\right). \tag{13}$$

In many case, the covariance matrix is dense while the precision matrix (or inverse covariance matrix) $\boldsymbol{\Gamma} = \boldsymbol{\Sigma}^{-1}$ is sparse (Fan et al., 2016), hence it is oftentimes economical-friendly to work with $\boldsymbol{\Gamma}$. We rewrite the block form distribution of spatiotemporal variables with precision matrix.

$$[\boldsymbol{x}, \boldsymbol{y}] \sim \mathcal{N}\left([\bar{\mathbf{x}}, \bar{\mathbf{y}}], \begin{bmatrix} \boldsymbol{\Gamma}_{\boldsymbol{xx}} & \boldsymbol{\Gamma}_{\boldsymbol{xy}} \\ \boldsymbol{\Gamma}_{\boldsymbol{yx}} & \boldsymbol{\Gamma}_{\boldsymbol{yy}} \end{bmatrix}^{-1}\right). \tag{14}$$

**Marginal distribution.** The marginal distribution of future temporal variable $\boldsymbol{y}$ is simply self-relevant from the mean and covariance

$$\boldsymbol{y} \sim \mathcal{N}\left(\bar{\mathbf{y}}, \boldsymbol{\Sigma}_{\boldsymbol{yy}}\right). \tag{15}$$

By the property of the inverse of block matrix (Choi, 2009), we have $\boldsymbol{\Sigma}_{\boldsymbol{yy}} = \left(\boldsymbol{\Gamma}_{\boldsymbol{yy}} - \boldsymbol{\Gamma}_{\boldsymbol{yx}}\boldsymbol{\Gamma}_{\boldsymbol{xx}}^{-1}\boldsymbol{\Gamma}_{\boldsymbol{xy}}\right)^{-1}$, hence we rewrite the marginal distribution of future temporal variable in the form of precision matrix,

$$\boldsymbol{y} \sim \mathcal{N}\left(\bar{\mathbf{y}}, \left(\boldsymbol{\Gamma}_{\boldsymbol{yy}} - \boldsymbol{\Gamma}_{\boldsymbol{yx}}\boldsymbol{\Gamma}_{\boldsymbol{xx}}^{-1}\boldsymbol{\Gamma}_{\boldsymbol{xy}}\right)^{-1}\right). \tag{16}$$

**Conditional distribution.** The conditional distribution of future temporal variable $\boldsymbol{y}$ with respect to history temporal variable $\boldsymbol{x} = \mathbf{x}$ is also a multivariate Gaussian distribution

$$\boldsymbol{y}|\boldsymbol{x} = \mathbf{x} \sim \mathcal{N}\left(\bar{\mathbf{y}} + \boldsymbol{\Sigma}_{\boldsymbol{yx}}\boldsymbol{\Sigma}_{\boldsymbol{xx}}^{-1}\left(\mathbf{x} - \bar{\mathbf{x}}\right), \boldsymbol{\Sigma}_{\boldsymbol{yy}} - \boldsymbol{\Sigma}_{\boldsymbol{yx}}\boldsymbol{\Sigma}_{\boldsymbol{xx}}^{-1}\boldsymbol{\Sigma}_{\boldsymbol{xy}}\right). \tag{17}$$

Moreover, one can show that $\boldsymbol{\Sigma}_{\boldsymbol{yx}}\boldsymbol{\Sigma}_{\boldsymbol{xx}}^{-1} = -\boldsymbol{\Gamma}_{\boldsymbol{yy}}^{-1}\boldsymbol{\Gamma}_{\boldsymbol{yx}}$ and $\left(\boldsymbol{\Sigma}_{\boldsymbol{yy}} - \boldsymbol{\Sigma}_{\boldsymbol{yx}}\boldsymbol{\Sigma}_{\boldsymbol{xx}}^{-1}\boldsymbol{\Sigma}_{\boldsymbol{xy}}\right) = \boldsymbol{\Gamma}_{\boldsymbol{yy}}^{-1}$ by the block matrix inversion, then the conditional distribution can be written as

$$\boldsymbol{y}|\boldsymbol{x} = \mathbf{x} \sim \mathcal{N}\left(\bar{\mathbf{y}} - \boldsymbol{\Gamma}_{\boldsymbol{yy}}^{-1}\boldsymbol{\Gamma}_{\boldsymbol{yx}}\left(\mathbf{x} - \bar{\mathbf{x}}\right), \boldsymbol{\Gamma}_{\boldsymbol{yy}}^{-1}\right). \tag{18}$$

**Statements about parameters $W$ and $\theta$.** The two parameters of GMRF: $W$ and $\theta$ represent the noise level in the spatiotemporal environment and homophily of nodes in the spatiotemporal graph respectively.

We expain the proposition of $W$ firstly. If we assume there is no correlation between nodes, i.e., $\mathcal{A}\left(\mathbf{A}\right) = \mathbf{0}$, then the potential matrix of MRF in the definition 4.1 reduces to

$$\boldsymbol{\Gamma} = W \otimes \mathbf{I}_N \in \mathbb{R}^{[(T+T_P)N] \times [(T+T_P)N]}. \tag{19}$$

Thus by the corresponding spatiotemporal multivariate gaussian distribution in the above explanation, as example of variable $\boldsymbol{y}|\boldsymbol{x} = \mathbf{x}$, the covariance matrix $\boldsymbol{\Sigma}'$ of multivariate random variable $\boldsymbol{y}_t|\boldsymbol{x} = \mathbf{x}$ for arbitrary $t = 1, 2, ..., T_P$ satisfying

$$\boldsymbol{\Sigma}' = \left(\boldsymbol{\Gamma}_{\boldsymbol{y}_t, \boldsymbol{y}_t}\right)^{-1} = \left(W_{t',t'}\mathbf{I}_N\right)^{-1} = W_{t',t'}^{-1}\mathbf{I}_N, \tag{20}$$

for $t' = T + t$. Hence variables in $\boldsymbol{y}_t|\boldsymbol{x} = \mathbf{x}$ are independent and identically distributed (i.i.d.) and variance of each variable in it is $W_{t',t'}^{-1}$, that is what we claim.

As to $\theta$, the greater the value of $\theta$ means that the feature of nodes in the corresponding time step is more compatible. We consider the extreme cases. If all entries in $\theta$ are 0, which reduce to the case deliberated above, then all nodes all i.i.d. through all time step, that is data among on the nodes is not circulating, which is the most heterogeneous situation. If all entries in $\theta$ is converge to positive infinity, then the data on the node is independent of itself and is only equal to the normalized summation of the adjacent node (Zhou et al., 2003).

## A.2 PROOFS OF THEORY

**Proof of Theory 1** By the definition 4.1 of GMRF, we can define the multivariate Gaussian distribution A.1 of probability density function,

$$f_{\boldsymbol{T}}(\mathbf{T}) = (2\pi)^{\frac{-N(T+T_P)}{2}} \det\left(\boldsymbol{\Gamma}^{-1}\right)^{\frac{1}{2}} \exp\left(-\frac{1}{2}\mathcal{V}(\mathbf{T})^{\top}\boldsymbol{\Gamma}\mathcal{V}(\mathbf{T})\right),  \tag{21}$$

where

$$\boldsymbol{\Gamma} = \begin{bmatrix} \boldsymbol{\Gamma}_{\boldsymbol{xx}} & \boldsymbol{\Gamma}_{\boldsymbol{xy}} \\ \boldsymbol{\Gamma}_{\boldsymbol{yx}} & \boldsymbol{\Gamma}_{\boldsymbol{yy}} \end{bmatrix} = \boldsymbol{\Sigma}^{-1},  \tag{22}$$

is the precision matrix, i.e., the inverse of covariance matrix $\boldsymbol{\Sigma}$. The temporal tensor are jointly sampled via multivariate Gaussian distribution $\mathcal{V}(\boldsymbol{T}) \sim \mathcal{N}(\mathbf{0}, \boldsymbol{\Gamma}^{-1})$. Here, $W$ satisfying symmetric positive definite and $\theta$ satisfying entry-wise positive are the pseudo parameters of standard MRF model. Hence we have $\boldsymbol{y}|\mathbf{x} \sim \mathcal{N}\left(-\boldsymbol{\Gamma}_{\boldsymbol{yy}}^{-1}\boldsymbol{\Gamma}_{\boldsymbol{yx}}\mathcal{V}(\mathbf{x}), \boldsymbol{\Gamma}_{\boldsymbol{yy}}^{-1}\right)$, i.e.,

$$\mathbb{E}[\boldsymbol{y}|\mathbf{x}] = -\boldsymbol{\Gamma}_{\boldsymbol{yy}}^{-1}\boldsymbol{\Gamma}_{\boldsymbol{yx}}\mathcal{V}(\mathbf{x}) = -\left(W_{\boldsymbol{yy}}\otimes\mathbf{I}_N + \mathrm{diag}\left(\theta_{\boldsymbol{y}}\right)\otimes\mathcal{A}(\mathbf{A})\right)^{-1}\left(W_{\boldsymbol{yx}}\otimes\mathbf{I}_N\right))\mathcal{V}(\mathbf{x}).  \tag{23}$$

Hence for arbitrary $t \in 1, 2, ..., T_P$, we have

$$\mathbb{E}[\boldsymbol{y}_t|\mathbf{x}] = -\left(W_{T+t,T+t}\mathbf{I}_N + \theta_{T+t}\mathcal{A}(\mathbf{A})\right)^{-1}\left(W_{T+t,1:T}\otimes\mathbf{I}_N\right)\mathcal{V}(\mathbf{x}),  \tag{24}$$

$$= -\left(W_{T+t,T+t}\mathbf{I}_N + \theta_{T+t}\mathcal{A}(\mathbf{A})\right)^{-1}\mathbf{x}^{\top}\times_2 W_{T+t,1:T}^{\top},  \tag{25}$$

$$= \left(W_{T+t,T+t}\mathbf{I}_N + \theta_{T+t}\mathcal{A}(\mathbf{A})\right)^{-1}\times_2\left(-W_{T+t,1:T}\mathbf{x}\right)^{\top}.  \tag{26}$$

where $\times_i$ is the matrix multiplication of tensor on the $i$-th dimension. Let $\alpha_t = \frac{\theta t}{W_{T+t,T+t}}$ and $\beta_t = -\frac{W_{T+t,1:T}}{W_{T+t,T+t}}$, we obtain reduce the above equation like

$$\mathbb{E}[\boldsymbol{y}_t|\mathbf{x}] = \left(\mathbf{I}_N + \alpha_t\mathcal{A}(\mathbf{A})\right)^{-1}\mathbf{x}^{\top}\times_2\boldsymbol{\beta}_t^{\top},  \tag{27}$$

$$= \left(\mathbf{I}_N + \alpha_t\mathcal{A}(\mathbf{A})\right)^{-1}\times_2\left(\boldsymbol{\beta}_t\mathbf{x}\right)^{\top}.  \tag{28}$$

Moreover, since $\lim_{k\to\infty}\mathcal{N}(A)^k = \mathbf{0}$, we expand $\left(\mathbf{I}_N + \alpha_t\mathcal{A}(\mathbf{A})\right)^{-1}$ in terms of the Neumann series (Moulinec et al., 2018) as

$$\left(\mathbf{I}_N + \alpha_t\mathcal{A}(\mathbf{A})\right)^{-1} = \left((1+\alpha_t)\mathbf{I}_N - \alpha_t\mathcal{N}(\mathbf{A})^{-1}\right),  \tag{29}$$

$$= \frac{1}{1+\alpha_t}\left(\mathbf{I}_N - \frac{\alpha_t}{1+\alpha_t}\mathcal{N}(\mathbf{A})\right)^{-1},  \tag{30}$$

$$= (1-\gamma_t)\sum_{k=0}^{\infty}\left(\gamma_t\mathcal{N}(\mathbf{A})\right)^k.  \tag{31}$$

where $\gamma_t = \alpha_t/(1+\alpha_t)$. Hence we can get

$$\mathbb{E}[\boldsymbol{y}_t|\mathbf{x}] = (1-\gamma_t)\sum_{k=0}^{\infty}\left(\gamma_t\mathcal{N}(\mathbf{A})\right)^k\mathbf{x}^{\top}\times_2\boldsymbol{\beta}_t^{\top},  \tag{32}$$

$$= (1-\gamma_t)\sum_{k=0}^{\infty}\left(\gamma_t\mathcal{N}(\mathbf{A})\right)^k\times_2\left(\boldsymbol{\beta}_t\mathbf{x}\right)^{\top}, \forall t \in 1, 2, ..., T_P,  \tag{33}$$

which completes the proof. $\qquad\square$

**Proof of Theory 2** Without loss of generality, we simplify the subsequent calculations by assuming two nodes disjoint union partition $V = V_1 \cup V_2$, i.e., $V_1 \cap V_2 = \emptyset$ to explore what are the implications for spatiotemporal learning when adding future impacts between data. Recall the result of Theory 1 and above proof, we get

$$\boldsymbol{y}|\mathbf{x} \sim \mathcal{N}\left(\mathbb{E}[\boldsymbol{y}|\mathbf{x}], \boldsymbol{\Gamma}_{\boldsymbol{yy}}^{-1}\right),  \tag{34}$$

hence the conditional distribution of $\boldsymbol{y}_{t,V_1}$ respect to $\mathbf{y}_{t,V_2}$ and $\mathbf{x}$ for the disjoint union $V_1 \cup V_2$ of node set $V$ and arbitrary $t = 1, 2, ..., T_P$ is

$$\boldsymbol{y}_{t,V_1}|\mathbf{x}, \mathbf{y}_{t,V_2} \sim \mathcal{N}\left(\mathbb{E}[\boldsymbol{y}_{t,V_1}|\mathbf{x}] + \boldsymbol{\Gamma}_{t,V_1V_1}^{-1}\boldsymbol{\Gamma}_{t,V_1V_2}\times_2\left(\mathbb{E}[\boldsymbol{y}_{t,V_1}|\mathbf{x}] - \mathbf{y}_{t,V_2}\right), \boldsymbol{\Gamma}_{t,V_1V_1}^{-1}\right),  \tag{35}$$

where $\mathbf{y}_{t,V_i} \coloneqq \left[\mathbf{y}_{t,u,:}^{\top} \mid \forall u \in V_i\right]^{\top}$ for $i = 1, 2$. Hence the above expectation is,

$$\mathbb{E}\left[\boldsymbol{y}_{t,V_1} | \mathbf{x}, \mathbf{y}_{t,V_2}\right] \tag{36}$$

$$= \mathbb{E}\left[\boldsymbol{y}_{t,V_1} | \mathbf{x}\right] + \boldsymbol{\Gamma}_{t,V_1 V_1}^{-1} \boldsymbol{\Gamma}_{t,V_1 V_2}\left(\mathbb{E}\left[\boldsymbol{y}_{t,V_1} | \mathbf{x}\right] - \mathbf{y}_{t,V_2}\right), \tag{37}$$

$$= \mathbb{E}\left[\boldsymbol{y}_{t,V_1} | \mathbf{x}\right] + \left(W_{T+t,T+t}\mathbf{I}_N + \theta_{T+t}\mathcal{A}\left(\mathbf{A}\right)\right)_{V_1 V_1}^{-1}\left(W_{T+t,T+t}\mathbf{I}_N + \theta_{T+t}\mathcal{A}\left(\mathbf{A}\right)\right)_{V_1 V_2} \times_2 \mathbf{r}_{t,V_2}, \tag{38}$$

$$= \mathbb{E}\left[\boldsymbol{y}_{t,V_1} | \mathbf{x}\right] + \left(\mathbf{I}_N + \alpha_t\mathcal{A}\left(\mathbf{A}\right)\right)_{V_1 V_1}^{-1}\left(\mathbf{I}_N + \alpha_t\mathcal{A}\left(\mathbf{A}\right)\right)_{V_1 V_2} \times_2 \mathbf{r}_{t,V_2}, \tag{39}$$

$$= \mathbb{E}\left[\boldsymbol{y}_{t,V_1} | \mathbf{x}\right] + \left(1 - \gamma_t\right)\sum_{k=0}^{\infty}\left(\gamma_t\mathcal{N}\left(\mathbf{A}\right)_{V_1, V_1}\right)^k\left(\mathbf{I}_N + \alpha_t\mathcal{A}\left(\mathbf{A}\right)\right)_{V_1, V_2} \times_2 \mathbf{r}_{t,V_2}, \tag{40}$$

still from the expansion of Neumann series (Moulinec et al., 2018) where $\alpha_t = \frac{\theta_t}{W_{T+t,T+t}}$ and $\gamma_t = \alpha_t / \left(1 + \alpha_t\right)$. The term $\left(\mathbf{I}_N + \alpha_t\mathcal{A}\left(\mathbf{A}\right)\right)_{V_1, V_2}$ indicates the submatrix consisting of rows corresponding to entries in $V_1$ and columns corresponding to entries in $V_2$ for $\mathbf{I}_N + \alpha_t\mathcal{A}\left(\mathbf{A}\right)$, similarity to $\mathcal{N}\left(\mathbf{A}\right)_{V_1, V_1}$, which illustrates the dynamics of residual propagation in this context. It must be noted, however, that the results of the closed form are independent of the node disjoint union partition chosen, as determined by the equivariance of the GMRF (Baz et al., 2022). Hence, the case we considered in Theory 2 is just a special example in the proof when $V_1 = \{u\}$ and $V_2 = V \setminus \{u\}$.□

### A.3 RESIDUAL PROPAGATION KERNEL

In traditional spatiotemporal graph learning, there are four widely used approaches to generate the associations between nodes (i.e., adjacency kernel function): predefined kernel, adaptive kernel, and data-driven kernel.

**Predefined kernel.** This kernel is typically constructed based on various prior information, such as the geographical information of nodes. This kernel function remains static during the model learning process. Specifically, for traffic data, we calculate the geographical distance between nodes $\mathbf{d}_s \in \mathbb{R}^{N \times N}$ (Li et al., 2017; Wu et al., 2019b; Yu et al., 2017; Liu et al., 2024b) (Shuman et al., 2013), then we construct the adjacency matrix kernel in the following manner:

$$\mathbf{A}_s \coloneqq e^{-\frac{\mathbf{d}_s}{\sigma^2}} \odot \mathbb{I}_{\{\mathbf{d}_s < -\sigma^2 \ln \varepsilon | \varepsilon \in (0,1)\}} \text{ and } \mathcal{N}\left(\mathbf{A}_s\right) = \mathbf{A}_s\mathbf{D}_s^{-1}, \tag{41}$$

with degree matrix $\mathbf{D}_s$ and diagonalization operator $\mathrm{diag}$. $\mathbb{I}$ is indicator function[1] and hyperparameter $\varepsilon \in (0, 1)$ filters through an extremely weak correlation to ease the burden of training. $\sigma$ is the standard deviation of $\mathbf{d}_s$. $\odot$ is Hadamard Product. And for atmosphere data, we calculate the geographic adjacency matrix based on longitude-latitude geodesic distance matrix $\mathbf{d}_{geo} \in \mathbb{R}^{N \times N}$ (Wang et al., 2014) and relative altitude matrix $\mathbf{h}_{alt} \in \mathbb{R}^{N \times N}$ (Wang et al., 2020) if existing,

$$\mathbf{A}_{geo} \coloneqq \mathbb{I}_{\{\mathbf{d}_{geo} < \varepsilon | \varepsilon > 0\}} \odot \mathbb{I}_{\{\mathbf{h}_{alt} < \xi | \xi > 0\}} \text{ and } \mathcal{N}\left(\mathbf{A}_{geo}\right) = \mathbf{D}_{geo}^{-1/2}\mathbf{A}_{geo}\mathbf{D}_{geo}^{-1/2}, \tag{42}$$

where $\mathbf{h}_{alt}\left[u, v\right] \coloneqq \sup_{\lambda \in (0,1)}\left\{h_w - \max\left\{h_u, h_v\right\} | w = \lambda u + (1 - \lambda) v\right\}$.

**Diffusion kernel.** The diffusion kernel represents a diffusion process where information is assumed to transfer from one node to its neighboring nodes with certain transition probabilities. This concept has a strong analogy in spatiotemporal graph domains, such as the traffic flow between nodes, which can be viewed as a diffusion process. Specifically, it is generally obtained in the following ways:

$$\mathbf{A}_{double} \coloneqq \left[\mathbf{A}_{road}, \mathbf{A}_{road}^{\top}\right] \text{ and } \mathcal{N}\left(\mathbf{A}_{double}\right) = \mathbf{A}_{double}\mathbf{D}_{double}^{-1}. \tag{43}$$

**Adaptive kernel.** The adaptive kernel is generated with two learnable node embeddings that can capture more complex node features from the data (Wu et al., 2019b; Shao et al., 2022d; Bai et al., 2020), which can be computed as:

$$\mathbf{A}_{adp} \coloneqq \mathrm{ReLU}\left(\mathbf{E}_1\mathbf{E}_2^{\top}\right) \text{ and } \mathcal{N}\left(\mathbf{A}_{adp}\right) = \mathrm{Softmax}\left(\mathbf{A}_{adp} - \mathrm{diag}\left(\mathbf{A}_{adp}\right)\right), \tag{44}$$

where $\mathbf{E}_1$ and $\mathbf{E}_2 \in \mathbb{R}^{N \times d_{adp}}$ are two learning node embeddings.

---

[1]The indicator function can also be replaced by some transformation of Heaviside step function (Weisstein, 2002).

**Data-driven kernel.** This kernel is generated by a complex neural network, notably using the Transformer architecture (Shao et al., 2022d; Jiang et al., 2023). Following these inspirations, we use Transformer to compute this kind of kernel:

$$\mathbf{A}_{att} := \frac{\mathrm{MLP}_1\left(\mathbf{Z}_{res}^{(t)}\right)\mathrm{MLP}_2\left(\mathbf{Z}_{res}^{(t)}\right)^\top}{\sqrt{d_{hid}}} \text{ and } \mathcal{N}\left(\mathbf{A}_{att}\right) = \mathrm{Softmax}\left(\mathbf{A}_{att} - \mathrm{diag}\left(\mathbf{A}_{att}\right)\right), \quad (45)$$

where $\mathbf{Z}_{res}^{(t)}$ is the residual representation of $t$-th time step.

### A.4  IMPORTANT VARIABLES AND DEFINITIONS

We explain the meaning or definition of each variable in detail, as shown in Table 6.

Table 6: Some important variables and their definitions.

| Variable | Definition |
|---|---|
| $\mathbf{x}/\boldsymbol{x}$ | Data/ its corresponding variable in GMRF |
| $\mathbf{y}/\boldsymbol{y}$ | Label/ its corresponding variable in GMRF |
| $W/\theta$ | Parameters of GMRF |
| $\mathbf{Z}_e$ | Input representation |
| $\mathbf{Z}_h$ | Label representation |
| $\mathbf{Z}_{res}/\mathbf{Z}_{res}$ | Residual representation/Smoothed residual representation |
| $y_{base}/y_{corr}$ | Base predition/Prediction correction |
| $\mathcal{F}_E$ | Input encoder |
| $\mathcal{F}_{ST}$ | STNN |
| $\mathcal{F}_D$ | Base encoder |
| $\mathcal{F}_R$ | Residual decoder |
| $T$ | The length of input time step |
| $N$ | The number of nodes |
| $T_p$ | The length of label time step |
| $f$ | The number of features of spatiotemporal data |
| $K$ | The number of kernels in residual propagation kernel |
| $L$ | The number of residual propagation layers |

### A.5  PSEUDOCODE OF STBIM

In Algorithm 1, we present the pseudocode of STBIM, including the forward learning process and the backward correction process. The forward spatiotemporal learning aims to capture the spatiotemporal features of input data and generate label representations. The backward correction process utilizes the inconsistencies between input and label representations to generate correction terms. It is worth noting that we use label representations instead of directly using labels, **eliminating the need for direct access to labels** in our method. Label representations can be seen as high-dimensional feature mappings of labels, preserving rich information.

## B  EXPERIMENTS

### B.1  DATASETS DETAILS

LargeST-SD, -GBA, -GLA, and -CA datasets used in our experiments are indeed subsets of the LargeST which is a large-scale traffic benchmark introduced in (Liu et al., 2023b). LargeST is a comprehensive dataset specifically designed for evaluating spatiotemporal traffic prediction tasks. It collects highway speed records from the PeMS (Performance Measurement System) with a sampling frequency of 15 minutes over a period of 5 years. In our experiments, we use LargeST-SD, -GBA, and -CA datasets in 2019. The datails are shown in Table 7.

The PEMS3-Stream dataset (Chen et al., 2021) is gathered by the California Transportation Agencies (CalTrans) Performance Measurement System (PeMS) in real-time at 30-second intervals. The data

---

**Algorithm 1:** STBIM for spatiotemporal prediction

---

**Input:** Input data $\mathbf{x} \in \mathbb{R}^{T \times N \times f}$ ;                                   // No label required
**Output:** Future label $\hat{\mathbf{y}} \in \mathbb{R}^{T_P \times N \times f}$
1  $\mathbf{Z}_e \leftarrow \mathcal{F}_E(\mathbf{x})$;                                   // Input representation
2  **# Forward spatiotemporal learning**;
3  $\mathbf{Z}_h \leftarrow \mathcal{F}_{ST}(\mathbf{Z}_e)$;                                   // Label representation learning
4  $\mathbf{y}_{base} \leftarrow \mathcal{F}_{ST}(\mathbf{Z}_h)$;                                   // Base prediction
5  **# Backward residual correction**;
6  $\mathbf{Z}_{res} \leftarrow \mathcal{F}_{ST}(\mathbf{Z}_e - \text{MLP}(\mathbf{Z}_h))$;                                   // Residual learning
7  $\tilde{\mathbf{Z}}_{res} = \left[ \boldsymbol{\tau}\left(\mathbf{I}_N + \frac{1}{K}\sum_{i=1}^{K}\boldsymbol{\alpha}_i \mathcal{K}_i\right) \times_2 \mathbf{Z}_{res} \right]^L$;                                   // Residual propagation
8  $\mathbf{y}_{corr} \leftarrow \mathcal{F}_R\left(\tilde{\mathbf{Z}}_{res}\right)$;                                   // Correction prediction
9  **# Final prediction**;
10  $\hat{\mathbf{y}} \leftarrow \mathbf{y}_{base} + \mathbf{y}_{corr}$;                                   // Final prediction

---

is aggregated into 5-minute intervals from the 30-second data instances. PEMS3-Stream comprises traffic flow data from 655 nodes in the North Central Area, collecting data for the month of July from 2011 to 2017, with a sampling frequency of 5 minutes. For our experiment, we utilized the data in 2011.

The KnowAir dataset (Wang et al., 2020) is a collection of PM2.5 measurements from 184 cities in China, covering a period of four years from January 1, 2015, to December 31, 2018. To comprehensively evaluate the capabilities of the models, the dataset is divided into three subsets along the time dimension, as presented in Table 8.

Table 7: The details of traffic datasets used in this paper.

| Dataset | Nodes | Edges | Time Range | Frames |
|---|---|---|---|---|
| LargeST-SD | 716 | 17,319 | 01/01/2019-31/12/2019 | 525,888 |
| LargeST-GBA | 2,352 | 61,246 | 01/01/2019-31/12/2019 | 525,888 |
| LargeST-GLA | 3,834 | 98,703 | 01/01/2019-31/12/2019 | 525,888 |
| LargeST-CA | 8,600 | 201,363 | 01/01/2019-31/12/2019 | 525,888 |
| PEMS03 | 358 | 546 | 09/01/2018 – 11/30/2018 | 26,208 |
| PEMS04 | 307 | 338 | 01/01/2018 – 02/28/2018 | 16,992 |
| PEMS08 | 170 | 276 | 07/01/2016 – 08/31/2016 | 17,856 |
| PEMS07 | 883 | 865 | 05/01/2017 – 08/06/2017 | 28,224 |
| METR-LA | 207 | 1,515 | 03/01/2012 – 06/27/2012 | 34,272 |
| PEMS3-Stream | 655 | 1,577 | 07/01/2011 - 07/31/2011 | 8,928 |

Table 8: The details of atmospheric datasets used in this paper.

| Dataset | KnowAir-1 | KnowAir-2 | KnowAir-3 |
|---|---|---|---|
| Nodes | 184 | 184 | 184 |
| Train range | 2015/1/1 - 2016/12/31 | 2015/11/1 - 2016/2/28 | 2016/9/1 - 2016/11/30 |
| Validate range | 2017/1/1 - 2017/12/31 | 2016/11/1 - 2017/2/28 | 2016/12/1 - 2016/12/31 |
| Test range | 2018/1/1 - 2018/12/31 | 2017/11/1 - 2018/2/28 | 2017/1/1 - 2017/1/31 |
| Sampling frequency | 3 hour | 3 hour | 3 hour |

## B.2 BASELINES

In this section, we describes the baselines used in detail. Most of the model codes with their hyperparameters are from the official benchmark LargeST (Liu et al., 2023b) and KownAir (Wang et al., 2020), with a small number of models sourced from their official codes.

- **HL** (Liang et al., 2021) selects the data from the last observation as the predicted value for all future time points.

- **LSTM** (Hochreiter and Schmidhuber, 1997) is an RNN variant to model long-term temporal dependencies.

- **STAEFormer** (Liu et al., 2023a) presents a novel component called spatiotemporal adaptive embedding that can yield outstanding results with vanilla transformers.

- **STGCN** (Yu et al., 2017) consists of multiple spatiotemporal convolution blocks, each of which forms a "sandwich" structure with two gated sequence convolution layers and a spatial graph convolution layer in the middle.

- **AGCRN** (Bai et al., 2020) proposes an adaptive graph convolution network to automatically capture fine-grained spatiotemporal correlations of traffic sequences.

- **DGCRN** (Li et al., 2023) uses the hypernetwork to exploit and extract the dynamic features of the node properties, while the parameters of the dynamic filter are generated at each time step.

- **DGCRN** (Weng et al., 2023) generates spatiotemporal embeddings using time information in traffic signals, and combines spatiotemporal embeddings with dynamic signals extracted from graph data to generate dynamic semantic graphs.

- **STID** (Shao et al., 2022b) is based on a fully connected layer architecture and incorporates additional spatiotemporal identity information to enhance performance.

- **STNN** (He et al., 2020b) is a spatiotemporal Transformer network model, which combines dynamic directed spatial dependence and long-term dependence to improve the accuracy of spatiotemporal graph prediction.

- **GC-LSTM** (Qi et al., 2019) integrates LSTM as the updating function and GCN to model the temporal and spatial dependency respectively.

- **PM2.5-GNN** (Wang et al., 2020) is knowledge-enhanced GNN devised to capture pollutants' horizontal transport by leveraging neighboring information and updating nodes' representations. A spatiotemporal GRU is applied after updates to model pollutants' vertical accumulation and diffusion under the influence of weather.

- **NodeFC-GRU** (Wang et al., 2020) is a degrading version of PM2.5GNN. It is implemented by replacing the GNN module in PM2.5-GNN with MLPs.

- **D$^2$STGNN** (Shao et al., 2022d) can decouple the hidden time series generated by the diffusion process from the hidden time series independent of other sensors, allowing for more accurate modeling of different parts of the traffic data.

- **GWNet** (Wu et al., 2019b) is based on a wavenet structure with double transition matrices, which are used to simulate the diffusion process of traffic flow. Furthermore, an adaptive matrix is employed to enhance the model.

- **STNorm** (Deng et al., 2021) is based on the Wavenet structure and uses the special spatial and temporal regularisation approach to complete the feature extraction of the spatiotemporal features.

- **stemGNN** (Cao et al., 2020) combines the Graph Fourier Transform (GFT) and the Discrete Fourier Transform (DFT), where GFT models inter-series correlations and DFT models temporal dependencies in an end-to-end framework.

- **STWA** (Cirstea et al., 2022) encodes time series from different locations into stochastic variables, from which we generate location-specific and time-varying model parameters to better capture the spatiotemporal dynamics.

Table 9: Predictive performance of the model on KnowAir dataset. 'Average improvement' reports the improvement of average prediction performance during 12 time steps using STRID relative to only baselines.

**KnowAir-1**

| Method | STBIM | Average | | | | | | Average relative improvement | | | | | |
|---|---|---|---|---|---|---|---|---|---|---|---|---|---|
| | | MAE ↓ | RMSE ↓ | MAPE (%) ↓ | CSI (%) ↑ | POD (%) ↑ | FAR (%) ↓ | MAE | RMSE | MAPE | CS | POD | FAR |
| nodesFC-GRU | - | 8.94 | 15.67 | 28.00 | 66.60 | 77.11 | 16.99 | - | - | - | - | - | - |
| | +JT | 8.91 | 15.65 | 27.21 | 66.36 | 76.58 | 16.75 | +0.34% | +0.13% | +2.82% | -0.36% | -0.69% | +1.41% |
| | +FT | 8.82 | 15.44 | 27.50 | 66.69 | 77.52 | 16.32 | +1.34% | +1.47% | +1.79% | +0.14% | +0.53% | +3.94% |
| GC_LSTM | - | 9.18 | 16.01 | 28.40 | 65.94 | 75.07 | 16.80 | - | - | - | - | - | - |
| | +JT | 8.82 | 15.45 | 27.36 | 66.92 | 75.71 | 14.77 | +3.92% | +3.50% | +3.66% | +1.49% | +0.85% | +12.08% |
| | +FT | 9.15 | 15.94 | 28.37 | 65.78 | 74.88 | 15.60 | +0.33% | +0.44% | +0.11% | -0.24% | -0.25% | +7.14% |
| STID | - | 7.95 | 13.99 | 25.22 | 70.66 | 78.16 | 14.36 | - | - | - | - | - | - |
| | +JT | 7.84 | 13.84 | 24.21 | 70.73 | 79.62 | 13.64 | +1.38% | +1.07% | +4.00% | +0.10% | +1.87% | +5.01% |
| | +FT | 7.85 | 13.89 | 23.60 | 70.47 | 78.84 | 13.10 | +1.26% | +0.71% | +6.42% | -0.27% | +0.87% | +8.77% |
| STAEFormer | - | 7.60 | 13.48 | 22.21 | 71.24 | 79.81 | 13.09 | - | - | - | - | - | - |
| | +JT | 7.55 | 13.36 | 22.17 | 71.54 | 79.91 | 12.76 | +0.66% | +0.89% | +0.18% | +0.42% | +0.13% | +2.52% |
| | +FT | 7.54 | 13.42 | 22.16 | 71.29 | 80.10 | 13.06 | +0.79% | +0.45% | +0.23% | +0.07% | +0.36% | +0.23% |
| AGCRN | - | 7.85 | 13.86 | 23.56 | 70.50 | 80.05 | 14.47 | - | - | - | - | - | - |
| | +JT | 7.79 | 13.22 | 23.01 | 70.88 | 80.99 | 14.31 | +0.76% | +4.62% | +2.33% | +0.54% | +0.17% | +1.11% |
| | +FT | 7.80 | 13.8 | 23.08 | 70.60 | 80.92 | 14.17 | +0.64% | +0.43% | +2.04% | +0.14% | +1.09% | +2.07% |
| DDGCRN | - | 8.01 | 14.11 | 24.85 | 70.11 | 79.87 | 14.84 | - | - | - | - | - | - |
| | +JT | 7.93 | 14.03 | 23.86 | 70.93 | 79.01 | 14.11 | +1.00% | +0.57% | +3.98 % | +1.17% | +1.08% | +4.92% |
| | +FT | 8.01 | 14.11 | 24.68 | 70.07 | 79.97 | 14.33 | +0.00% | +0.00% | +0.68% | -0.06% | +0.13% | +3.44% |
| PM2.5GNN | - | 8.87 | 15.50 | 28.63 | 67.00 | 76.04 | 15.07 | - | - | - | - | - | - |
| | +JT | 8.49 | 15.02 | 24.66 | 67.50 | 75.43 | 13.47 | +4.28% | +3.10% | +13.87% | +0.75% | +0.80% | +10.62% |
| | +FT | 8.78 | 15.39 | 27.02 | 67.29 | 76.79 | 15.02 | +1.01% | +0.71% | +5.62% | +0.43% | +0.99% | +0.33% |

**KnowAir-2**

| Method | STBIM | Average | | | | | | Average relative improvement | | | | | |
|---|---|---|---|---|---|---|---|---|---|---|---|---|---|
| | | MAE ↓ | RMSE ↓ | MAPE (%) ↓ | CSI (%) ↑ | POD (%) ↑ | FAR (%) ↓ | MAE | RMSE | MAPE | CSI | POD | FAR |
| nodesFC-GRU | - | 14.28 | 24.82 | 31.53 | 69.77 | 80.58 | 16.13 | - | - | - | - | - | - |
| | +JT | 14.26 | 24.12 | 30.87 | 70.60 | 81.23 | 16.06 | +0.14% | +2.82% | +2.09% | +1.19% | +0.81% | +0.43% |
| | +FT | 14.27 | 24.66 | 31.46 | 69.97 | 81.85 | 16.15 | +0.07% | +0.64% | +0.22% | +0.29% | +1.58% | -0.12% |
| GC_LSTM | - | 14.87 | 25.71 | 33.05 | 68.36 | 80.22 | 17.79 | - | - | - | - | - | - |
| | +JT | 14.47 | 25.19 | 31.00 | 68.32 | 81.60 | 15.05 | +2.69% | +2.02% | +6.20% | -0.06% | +1.72% | +15.4% |
| | +FT | 14.08 | 25.99 | 33.01 | 68.39 | 81.02 | 16.84 | +5.31% | -1.09% | +0.12% | +0.04% | +1.00% | +5.34% |
| STID | - | 13.57 | 23.46 | 31.96 | 71.54 | 82.52 | 15.69 | - | - | - | - | - | - |
| | +JT | 13.19 | 22.94 | 29.35 | 72.45 | 83.71 | 15.66 | +2.80% | +2.22% | +8.17% | +1.27% | +1.44% | +0.19% |
| | +FT | 13.56 | 23.33 | 31.45 | 71.39 | 82.41 | 15.78 | +0.07% | +0.55% | +1.60% | -0.21% | -0.13% | -0.57% |
| STAEFormer | - | 13.21 | 23.65 | 29.90 | 72.70 | 83.30 | 16.89 | - | - | - | - | - | - |
| | +JT | 13.21 | 23.00 | 28.94 | 72.08 | 82.95 | 15.38 | +0.00% | +2.75% | +3.21% | -0.85% | -0.42% | +8.94% |
| | +FT | 13.09 | 22.71 | 29.90 | 72.87 | 84.86 | 16.24 | +0.91% | +0.39% | +0.00% | +0.23% | +1.87 | +3.85% |
| AGCRN | - | 13.88 | 24.24 | 30.10 | 70.24 | 79.28 | 13.97 | - | - | - | - | - | - |
| | +JT | 13.25 | 23.70 | 29.16 | 70.72 | 79.45 | 13.94 | +4.54% | +2.23% | +3.12% | + 0.68% | +0.21% | +0.21% |
| | +FT | 13.11 | 23.43 | 29.79 | 70.52 | 81.37 | 13.89 | +5.55% | +3.34% | +1.03% | +0.40% | +2.64% | +0.57% |
| DDGCRN | - | 13.99 | 24.16 | 33.48 | 70.83 | 81.33 | 16.47 | - | - | - | - | - | - |
| | +JT | 13.91 | 24.10 | 32.02 | 70.22 | 82.21 | 15.07 | +0.57% | +0.25% | +4.36% | -0.86% | +1.08% | +8.50% |
| | +FT | 13.94 | 24.13 | 32.45 | 70.66 | 82.09 | 15.40 | +0.36% | +0.12% | +3.08% | -0.24% | +0.93% | +6.50% |
| PM2.5GNN | - | 14.55 | 25.09 | 33.26 | 68.94 | 81.28 | 18.05 | - | - | - | - | - | - |
| | +JT | 14.39 | 24.90 | 32.39 | 69.20 | 81.21 | 17.61 | +1.01% | +0.76% | +2.62% | +0.38% | +0.09% | +2.44% |
| | +FT | 14.57 | 25.12 | 33.31 | 68.81 | 81.09 | 18.03 | -0.13% | -0.12% | -0.15% | -0.19% | -0.22% | +0.11% |

**KnowAir-3**

| Method | STBIM | Average | | | | | | Average relative improvement | | | | | |
|---|---|---|---|---|---|---|---|---|---|---|---|---|---|
| | | MAE ↓ | RMSE ↓ | MAPE(%) ↓ | CSI (%) ↑ | POD (%) ↑ | FAR (%) ↓ | MAE | RMSE | MAPE | CSI | POD | FAR |
| nodesFC-GRU | - | 20.48 | 35.10 | 39.11 | 71.54 | 87.88 | 21.42 | - | - | - | - | - | - |
| | +JT | 19.60 | 34.71 | 37.97 | 72.46 | 88.92 | 20.27 | +4.30% | +1.11% | +2.91% | +1.29% | +1.18% | +5.37% |
| | +FT | 19.88 | 34.63 | 35.63 | 72.62 | 87.42 | 18.90 | +2.93% | +1.34% | +8.90% | +1.51% | -0.52% | +11.76% |
| GC_LSTM | - | 20.91 | 36.00 | 39.32 | 72.30 | 88.94 | 20.56 | - | - | - | - | - | - |
| | +JT | 19.49 | 33.89 | 36.14 | 73.71 | 88.36 | 18.36 | +6.79% | +5.86% | +8.09% | +1.95% | -0.65% | +10.7% |
| | +FT | 20.86 | 35.94 | 39.05 | 72.34 | 88.61 | 20.37 | +0.24% | +0.17% | +0.69% | +0.06% | -0.37% | +0.92% |
| STID | - | 17.50 | 31.63 | 33.04 | 76.04 | 88.75 | 15.85 | - | - | - | - | - | - |
| | +JT | 17.47 | 30.88 | 32.07 | 76.60 | 92.13 | 15.33 | +0.17% | +2.37% | +2.94% | +0.74% | +3.81% | +3.28% |
| | +FT | 17.41 | 30.97 | 33.16 | 76.72 | 89.26 | 15.69 | +0.51% | +2.09% | -0.36% | +0.89% | +0.57% | +1.01% |
| STAEFormer | - | 18.28 | 31.33 | 36.42 | 75.68 | 90.89 | 19.67 | - | - | - | - | - | - |
| | +JT | 17.56 | 30.41 | 33.97 | 76.54 | 92.28 | 18.22 | +3.94% | +2.94% | +6.72% | +1.14% | +1.53% | +7.37% |
| | +FT | 17.64 | 30.70 | 32.57 | 76.25 | 90.63 | 17.22 | +3.50% | +2.01% | +10.57% | +0.75% | -0.29% | +12.46% |
| AGCRN | - | 19.83 | 34.25 | 39.00 | 73.24 | 90.14 | 20.38 | - | - | - | - | - | - |
| | +JT | 19.61 | 33.41 | 38.62 | 73.32 | 90.67 | 19.49 | +1.11% | +2.45% | +0.97% | +0.11% | +0.59% | +4.37% |
| | +FT | 19.66 | 34.17 | 36.95 | 73.19 | 90.77 | 19.34 | +0.86% | +0.23% | +5.26% | -0.07% | +0.70% | +5.10% |
| DDGCRN | - | 19.68 | 34.60 | 36.36 | 72.94 | 86.29 | 18.49 | - | - | - | - | - | - |
| | +JT | 19.00 | 34.22 | 36.28 | 72.91 | 88.54 | 17.86 | +3.46% | +1.01% | +0.22% | -0.04% | +2.61% | +3.41% |
| | +FT | 19.65 | 34.51 | 36.22 | 72.98 | 87.37 | 17.66 | +0.15% | +0.26% | +0.39% | +0.05% | +1.25% | +4.49% |
| PM2.5GNN | - | 20.28 | 34.97 | 38.25 | 72.27 | 88.43 | 20.18 | - | - | - | - | - | - |
| | +JT | 18.99 | 32.97 | 36.75 | 74.35 | 89.45 | 18.50 | +6.36% | +5.72% | +3.92% | +2.88% | +1.15% | +8.33% |
| | +FT | 19.39 | 33.69 | 35.43 | 73.57 | 88.19 | 18.39 | +4.39% | +3.66% | +7.73% | +1.80% | -0.27% | +8.87% |

### B.3 METRIC

To assess the efficacy of our framework, we employed metrics commonly utilized in spatiotemporal prediction tasks, including Mean Absolute Error (MAE), Root Mean Square Error (RMSE), and Mean Absolute Percentage Error (MAPE). Moreover, we consider specific metrics, including Critical Success Index (CSI), Probability of Detection (POD), and False Alarm Rate (FAR), to assess the performance of the system in atmospheric tasks. Let the prediction value be $\hat{\mathbf{y}}_{:,u}$ and ground truth value be $\mathbf{y}_{:,u}$ for a specific node $u$, then the common metrics satisfy,

$$\text{MAE} = \frac{\sum_{t=1}^{T_P} |\mathbf{y}_{t,u} - \hat{\mathbf{y}}_{t,u}|}{T_P}, \tag{46}$$

$$\text{RMSE} = \sqrt{\frac{\sum_{t=1}^{T_P} (\mathbf{y}_{t,u} - \hat{\mathbf{y}}_{t,u})^2}{T_P}}, \tag{47}$$

$$\text{MAPE} = \frac{1}{T_P} \sum_{t=1}^{T_P} \frac{|\mathbf{y}_{t,u} - \hat{\mathbf{y}}_{t,u}|}{\mathbf{y}_{t,u}}. \tag{48}$$

More over, we choose $\varepsilon = 75 \mu g/m^3$ to be the demarcation point of good air quality (Zhao et al., 2016). Hence the specific metrics satisfy,

$$\text{CSI} = \frac{\# \{t \mid \mathbf{y}_{t,u} \geq \varepsilon, \hat{\mathbf{y}}_{t,u} \geq \varepsilon\}}{24 - \# \{t \mid \mathbf{y}_{t,u} < \varepsilon, \hat{\mathbf{y}}_{t,u} < \varepsilon\}}, \tag{49}$$

$$\text{POD} = \frac{\# \{t \mid \mathbf{y}_{t,u} \geq \varepsilon, \hat{\mathbf{y}}_{t,u} \geq \varepsilon\}}{\# \{t \mid \mathbf{y}_{t,u} \geq \varepsilon, \hat{\mathbf{y}}_{t,u} \geq \varepsilon\} + \# \{t \mid \mathbf{y}_{t,u} \geq \varepsilon, \hat{\mathbf{y}}_{t,u} < \varepsilon\}}, \tag{50}$$

$$\text{FAR} = \frac{\# \{t \mid \mathbf{y}_{t,u} < \varepsilon, \hat{\mathbf{y}}_{t,u} \geq \varepsilon\}}{\# \{t \mid \mathbf{y}_{t,u} \geq \varepsilon, \hat{\mathbf{y}}_{t,u} \geq \varepsilon\} + \# \{t \mid \mathbf{y}_{t,u} < \varepsilon, \hat{\mathbf{y}}_{t,u} \geq \varepsilon\}}, \tag{51}$$

where $\#$ calculates the cardinal of the following set. It is important to note that smaller metrics represent better model performance for all metrics except CSI and POD. the opposite is true for CSI and POD metrics. It is crucial to acknowledge that smaller metrics indicate superior model performance for all metrics except CSI and POD. Conversely, the opposite is true for CSI and POD metrics.

### B.4 ANALYSIS OF EXPERIMENTAL RESULTS ON THE OTHER DATASETS

In this section, we analyze the effectiveness of the proposed module on the other traffic dataset and Know Air dataset from the atmospheric domain. Traffic datasets include PeMS03, PeMS04, PeMS08, PeMS07, METR-LA, PEMS3-Stream, Large-LA, and Large-CA.

For KnowAir datasets, we we also complement specialized models that perform well in atmospheric prediction tasks including GC-LSTM (Qi et al., 2019), nodesFC-GRU, and PM2.5GNN (Wang et al., 2020). In addition, for a more comprehensive evaluation, we introduce several indicators that are widely used in the field of atmospheric forecasting, including the critical success index (CSI), probability of detection (POD), and false alarm rate (FAR). Please mote that higher values for the first two metrics mean better performance. As shown in Table 9, we find that STID still performs the best on the KnowAir dataset, surpassing several dedicated atmospheric prediction models. This is because the fundamental challenge in both spatiotemporal atmospheric prediction and traffic tasks lies in modeling spatiotemporal correlations, which STID evidently does better. The experimental conclusions are consistent with those from the main experiments; our module can significantly aid spatiotemporal graph models in predicting future atmospheric data.

The results are shown in Table 10, and we can see that AGCRN exhibits low prediction errors due to its adaptive graph learning strategy, which enables more accurate capture of spatial dependencies. Interestingly, the MLP-based architecture STID demonstrates competitive performance, likely owing to its use of spatial identity encoding, enhancing the spatial representation of nodes. When the spatiotemporal prediction baselines are integrated with the proposed STBIM module, all models exhibit performance improvements. For complex models with numerous parameters, such as D$^2$STGNN, the proposed modules can enhance their representational capacity for inconsistencies. Especially on the PEMS3-Stream dataset, the limited amount of data for just one month results in D$^2$STGNN struggling

Table 10: Average performance of models on 8 datasets. "+STBIM" means the baseline with STBIM in the joint training manner. "Imp." is the percentage improvement of performance over the baseline. We bold the best performance in every baseline experiment.

| Dataset | PeMS03 | | | PeMS04 | | | PeMS08 | | | METR-LA | | |
|---|---|---|---|---|---|---|---|---|---|---|---|---|
| Method | MAE | RMSE | MAPE | MAE | RMSE | MAPE | MAE | RMSE | MAPE | MAE | RMSE | MAPE |
| LSTM | 21.33 | 35.11 | 23.33 | 23.81 | 36.62 | 18.12 | 21.31 | 32.10 | 17.47 | 3.55 | 7.10 | 10.18 |
| +STBIM | **16.21** | **27.44** | **15.83** | **21.37** | **33.65** | **14.30** | **16.27** | **26.04** | **10.43** | **3.27** | **6.42** | **9.28** |
| Imp. | +24.00% | +21.84% | +32.14% | +10.24% | +8.11% | +21.08% | +23.65% | +18.87% | +40.29% | +7.88% | +9.57% | +8.84% |
| STID | 15.36 | 25.97 | 16.20 | 18.60 | 30.14 | 12.28 | 14.21 | 23.43 | 9.28 | 3.22 | 6.58 | 9.16 |
| +STBIM | **15.08** | **25.75** | **15.49** | **17.89** | **29.66** | **11.99** | **13.80** | **23.09** | **9.18** | **3.06** | **6.19** | **8.54** |
| Imp. | +1.82% | +0.85% | +4.38% | +3.81% | +1.59% | +2.36% | +2.88% | +1.45% | +1.07% | +4.96% | +5.92% | +6.76% |
| STAEFormer | 15.45 | 27.39 | 15.08 | 18.17 | 29.99 | 11.92 | 13.59 | 23.93 | 8.83 | 3.02 | 6.07 | 8.35 |
| +STBIM | **15.13** | **26.80** | **14.72** | **18.02** | **28.56** | **11.08** | **13.36** | **23.35** | **8.43** | **2.99** | **6.06** | **8.16** |
| Imp. | +2.07% | +2.15% | +2.38% | +0.82% | +4.76% | +7.04% | +1.69% | +2.42% | +4.53% | +1.00% | +0.16% | +2.27% |
| STGCN | 17.47 | 28.81 | 17.08 | 20.01 | 31.82 | 13.32 | 15.69 | 25.19 | 10.31 | 3.11 | 6.26 | 8.60 |
| +STBIM | **16.74** | **28.12** | **16.89** | **19.13** | **30.83** | **13.15** | **15.07** | **24.50** | **9.91** | **3.03** | **6.10** | **8.24** |
| Imp. | +4.18% | +2.39% | +1.11% | +4.39% | +3.11% | +1.27% | +3.95% | +2.73% | +3.87% | +2.57% | +2.55% | +4.18% |
| AGCRN | 16.06 | 28.49 | 15.85 | 19.83 | 32.26 | 12.97 | 15.59 | 25.07 | 10.19 | 3.15 | 6.38 | 8.81 |
| +STBIM | **15.27** | **26.91** | **14.71** | **18.85** | **30.95** | **12.41** | **15.15** | **24.89** | **10.07** | **3.14** | **6.31** | **8.71** |
| Imp. | +4.91% | +5.54% | +7.19% | +4.94% | +4.06% | +4.31% | +2.82% | +0.71% | +1.17% | +0.31% | +1.09% | +1.13% |
| STNorm | 15.28 | 25.73 | 14.71 | 19.57 | 32.36 | 12.28 | 15.61 | 24.97 | 10.05 | 3.13 | 6.41 | 8.72 |
| +STBIM | **14.99** | **25.46** | **14.21** | **18.69** | **30.34** | **12.06** | **14.85** | **23.80** | **9.30** | **3.12** | **6.39** | **8.71** |
| Imp. | +1.89% | +1.05% | +3.40% | +4.49% | +6.42% | +1.79% | +4.86% | +4.68% | +7.46% | +0.32% | +0.31% | +0.11% |
| GWNet | 16.77 | 27.57 | 16.11 | 21.79 | 33.79 | 14.85 | 18.03 | 27.86 | 9.41 | 3.03 | 6.04 | 8.22 |
| +STBIM | **16.41** | **27.00** | **15.20** | **21.11** | **32.87** | **14.37** | **17.96** | **27.56** | **9.10** | **3.01** | **6.02** | **8.21** |
| Imp. | +2.14% | +2.06% | +5.64% | +3.12% | +2.72% | +3.23% | +0.38% | +1.07% | +3.29% | +0.66% | +0.33% | +0.12% |
| STWA | 15.19 | 26.76 | 15.99 | 19.37 | 31.28 | 12.63 | 15.59 | 24.67 | 10.79 | 3.30 | 6.71 | 9.45 |
| +STBIM | **14.96** | **25.80** | **15.66** | **18.90** | **30.50** | **12.01** | **14.95** | **23.86** | **10.77** | **3.29** | **6.66** | **9.24** |
| Imp. | +1.51% | +3.59% | +2.06% | +2.43% | +2.49% | +4.90% | +4.10% | +3.28% | +0.19% | +0.31% | +0.75% | +2.22% |
| stemGNN | 16.42 | 27.52 | 15.65 | 22.02 | 34.24 | 15.51 | 17.70 | 27.48 | 11.66 | 3.28 | 6.73 | 9.30 |
| +STBIM | **16.20** | **26.49** | **15.35** | **21.43** | **33.36** | **14.99** | **16.92** | **26.39** | **11.16** | **3.06** | **6.46** | **8.93** |
| Imp. | +1.33% | +3.74% | +1.92% | +2.68% | +2.57% | +3.35% | +17.65% | +4.40% | +3.96% | +6.70% | +4.01% | +3.97% |
| D²STGNN | 14.62 | 25.09 | 14.23 | 18.53 | 30.68 | 12.17 | 14.36 | 23.76 | 9.37 | 3.01 | 6.05 | 8.41 |
| +STBIM | **14.51** | **24.54** | **13.92** | **18.22** | **30.17** | **12.00** | **13.77** | **23.35** | **8.99** | **2.94** | **6.02** | **8.12** |
| Imp. | +0.75% | +1.35% | +2.14% | +1.67% | +1.66% | +1.40% | +4.10% | +1.73% | +4.05% | +1.00% | +0.50% | +3.45% |
| Dataset | PeMS07 | | | PEMS3-Stream | | | GLA | | | CA | | |
| Method | MAE | RMSE | MAPE | MAE | RMSE | MAPE | MAE | RMSE | MAPE | MAE | RMSE | MAPE |
| STNorm | 20.56 | 34.88 | 8.63 | 11.92 | 18.56 | 15.63 | 21.31 | 34.53 | 14.06 | 19.30 | 31.98 | 14.02 |
| +STBIM | **19.88** | **33.25** | **8.30** | **11.63** | **18.14** | 15.26 | **21.09** | **33.80** | 12.71 | **19.03** | **31.35** | **13.23** |
| Imp. | +3.30% | +4.67% | +3.82% | +2.43% | +2.37% | +1.29% | +1.03% | +2.11% | +9.60% | +1.40% | +1.94% | +5.62% |
| GWNet | 24.55 | 38.36 | 10.15 | 12.44 | 18.98 | 16.78 | 21.21 | 33.63 | 13.73 | 21.74 | 34.22 | 17.41 |
| +STBIM | **23.46** | **37.65** | **9.86** | **11.59** | **17.81** | 15.33 | **20.65** | **32.97** | **13.42** | **19.97** | **32.26** | **14.28** |
| Imp. | +4.44% | +1.85% | +2.85% | +3.74% | +3.56% | +8.64% | +3.60% | +1.96% | +2.25% | +8.14% | +3.37% | +3.13% |
| STID | 19.52 | 32.90 | 8.27 | 12.58 | 19.36 | 16.21 | 21.69 | 35.20 | 14.39 | 19.10 | 32.00 | 14.73 |
| +STBIM | **19.17** | **32.45** | **8.18** | **11.72** | **17.96** | 15.57 | **21.46** | 34.57 | **13.61** | **18.50** | **30.92** | **13.61** |
| Imp. | +1.79% | +1.37% | +1.09% | +6.83% | +7.23% | +3.95% | +1.06% | +1.78% | +5.28% | +3.14% | +3.38% | +7.60% |
| STGCN | 21.62 | 34.89 | 13.99 | 13.42 | 20.27 | 17.73 | 22.62 | 38.71 | 14.12 | 21.36 | 36.42 | 16.55 |
| +STBIM | **20.47** | **32.76** | **12.94** | **12.07** | **19.14** | 16.56 | **21.51** | **37.15** | **13.19** | **19.85** | **34.30** | **14.43** |
| Imp. | +10.05% | +5.57% | +6.60% | +5.73% | +4.91% | +4.03% | +6.59% | +2.02% | +0.35% | +7.06% | +5.82% | +12.80% |
| D²STGNN | 19.77 | 33.08 | 8.40 | 12.98 | 20.36 | 17.24 | | | | | | |
| +STBIM | **19.52** | **32.53** | **8.11** | **11.72** | **17.96** | 16.21 | | Out of memory | | | | |
| Imp. | +1.26% | +1.66% | +3.45% | +9.71% | +11.78% | +6.08% | | | | | | |

to fully learn spatiotemporal features, leading to an issue of underfitting. Our module serves as a powerful auxiliary tool to assist D$^2$STGNN in acquiring more information, thereby achieving a significant performance boost. Large-CA is a large-scale traffic dataset with 8600 nodes, which to our knowledge is currently the largest publicly available road network dataset. Even on large-scale datasets, our model continues to improve effectiveness of STNNs.

## B.5 Effectiveness of input-laebl inconsistent feature modeling

We evaluate the effectiveness of the model on samples with inconsistencies in spatial and temporal dimensions. We include more time-inconsistent samples by considering samples where the Surge/Plumment ratio of the mean of input data relative to the mean of label data ranged from 25% to 75% as time-inconsistent samples. As shown in Table 11 of the experimental results, we found that our proposed modules effectively enhance the modeling capability of STNN for spatiotemporal inconsistency features. For STID and D$^2$STGNN, the proposed modules explicitly utilize label features to mitigate the negative impact of these inconsistencies.

Table 11: Modeling preformance of temporal historical-future inconsistencies with different change radio. We use LargeST-SD dataset as example and train STBIM with STNNs in a join-training manner.

| Surge | | 25% - 50% | | | 50% - 75% | | | 75% - 100% | | |
|---|---|---|---|---|---|---|---|---|---|---|
| | | MAE | RMSE | MAPE | MAE | RMSE | MAPE | MAE | RMSE | MAPE |
| STGCN | - | 18.12 | 46.81 | 12.10 | 23.80 | 36.83 | 13.65 | 27.67 | 40.42 | 45.20 |
| | +STBIM | **15.07** | **29.82** | **10.32** | **18.13** | **27.55** | **13.06** | **24.34** | **37.60** | **42.27** |
| | Imp. | +16.83% | +36.29% | +14.71% | +23.82 | +25.19% | +4.32% | +12.03% | +6.98% | +6.48% |
| STID | - | 14.82 | 29.17 | 11.78 | 19.65 | 30.24 | 13.47 | 27.06 | 41.40 | 43.63 |
| | +STBIM | **12.04** | **26.12** | **9.16** | **15.15** | **24.02** | **10.56** | **20.55** | **32.16** | **33.50** |
| | Imp. | +18.76% | +10.46% | +22.24% | +22.90% | +20.57% | +21.60% | +24.06% | +22.32% | +23.21% |
| STAEformer | - | 12.79 | 25.57 | 10.21 | 15.26 | 28.61 | 11.64 | 25.63 | 36.06 | 35.26 |
| | +STBIM | **11.74** | **25.38** | **9.86** | **15.59** | **24.21** | **10.78** | **21.79** | **34.19** | **33.91** |
| | Imp. | +8.21% | +0.74% | +3.43% | 2.16% | +15.38% | +7.39% | +14.98% | +5.19% | +3.83% |
| D$^2$STGNN | - | 11.79 | 25.07 | 9.92 | 14.89 | 23.35 | 10.14 | 21.09 | 33.37 | 34.64 |
| | +STBIM | **11.38** | **24.40** | **9.04** | **14.25** | **22.59** | **9.98** | **20.31** | **31.37** | **32.85** |
| | Imp. | +3.48% | +2.67% | +8.87% | +4.30% | +3.25% | +1.58% | +3.70% | +5.99% | +5.17% |
| Plumment | | 25% - 50% | | | 50% - 75% | | | 75% - 100% | | |
| | | MAE | RMSE | MAPE | MAE | RMSE | MAPE | MAE | RMSE | MAPE |
| STGCN | - | 24.07 | 56.63 | 18.31 | 28.10 | 48.14 | 20.65 | 33.25 | 47.47 | 22.61 |
| | +STBIM | **19.59** | **35.74** | **13.06** | **21.03** | **36.49** | **16.81** | **22.95** | **39.08** | **19.73** |
| | Imp. | +18.61% | +36.89% | +28.67% | +25.16% | +24.20% | +18.60% | +30.98% | +17.67% | +12.74% |
| STID | - | 26.70 | 53.64 | 15.09 | 27.15 | 51.77 | 20.77 | 27.54 | 47.78 | 26.23 |
| | +STBIM | **14.44** | **26.78** | **9.50** | **15.38** | **27.26** | **11.92** | **17.18** | **29.59** | **14.24** |
| | Imp. | +45.92% | +50.07% | +37.04% | +43.35% | +47.34% | +42.61% | +37.62% | +38.07% | +45.71% |
| STAEformer | - | 18.12 | 37.22 | 13.25 | 19.94 | 37.92 | 26.00 | 23.40 | 39.45 | 18.62 |
| | +STBIM | **17.30** | **36.17** | **10.79** | **18.48** | **36.27** | **15.59** | **20.04** | **36.41** | **16.64** |
| | Imp. | +4.53% | +2.82% | +18.57% | +7.32% | +4.35% | +40.04% | +14.36% | +7.71% | +10.63% |
| D$^2$STGNN | - | 14.05 | 27.23 | 10.41 | 15.30 | 28.23 | 13.16 | 17.46 | 30.28 | 15.35 |
| | +STBIM | **13.54** | **26.58** | **9.61** | **15.11** | **27.44** | **11.89** | **17.09** | **29.78** | **14.06** |
| | Imp. | +3.63% | +2.39% | +7.68% | +1.24% | +2.80% | +9.65% | +2.12% | +1.65% | +8.40% |

## B.6 Comparison between STID and STBIM for input-laebl inconsistent modeling

**First and foremost, we emphasize that our motivation is not to surpass specific techniques within the model. Our contribution lies in introducing a general module to enhance existing spatiotemporal prediction models, which is orthogonal to existing technologies**. Just as the analysis below illustrates: the combination of STID and STBIM outperforms other variants in terms of inconsistent feature performance, our model can synergize with other advanced technologies to generate a broader and more comprehensive impact.

### B.6.1 PERFORMANCE COMPARISON

STID identifies spatiotemporal deviations and rectifies spatial inconsistencies using node embedding techniques. We integrated the embedding technique from STID with STBIM through a joint training approach. A comparison of spatial and temporal inconsistency samples is presented in Table 12.

Our analysis demonstrates that the proposed module excels in capturing spatiotemporal deviation features compared to node embedding techniques, particularly in addressing temporal inconsistency challenges. This enhanced performance is attributed to our method's explicit utilization of label information, resulting in more precise modeling. Additionally, it highlights that the node embedding technique introduced by STID may not effectively resolve spatiotemporal deviation issues. Conversely, STID+STBIM achieves competitive performance, showcasing the synergistic potential of integrating advanced technologies with STBIM.

Table 12: Modeling preformance of temporal historical-future inconsistencies with different change radio, and we use LargeST-SD dataset as example.

|  | Model | 25% - 50% | | | 50% - 75% | | | 75% - 100% | | |
|---|---|---|---|---|---|---|---|---|---|---|
|  |  | MAE | RMSE | MAPE | MAE | RMSE | MAPE | MAE | RMSE | MAPE |
| STID | - | 14.82 | 29.17 | 11.78 | 19.65 | 30.24 | 13.47 | 27.06 | 41.40 | 43.63 |
|  | w/o em+STBIM | 14.03 | 28.79 | 11.53 | 19.31 | 29.72 | 12.46 | 25.68 | 35.97 | 38.42 |
|  | STID+STBIM | 12.04 | 26.12 | 9.16 | 15.15 | 24.02 | 10.56 | 20.55 | 32.16 | 33.50 |

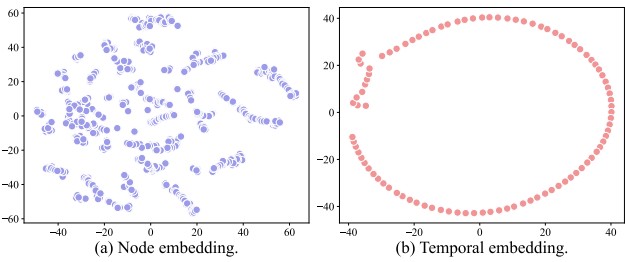

(a) Node embedding.    (b) Temporal embedding.

Figure 5: Node embedding and temporal embedding of STID in LargeST-SD dataset.

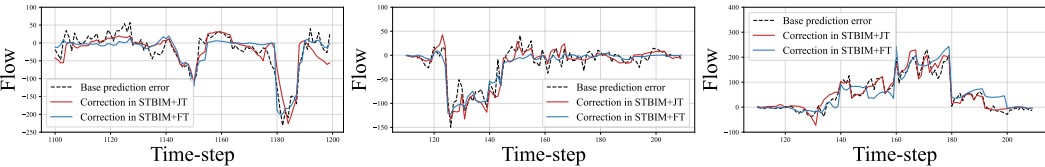

Figure 6: The prediction errors between base prediction and labels with correction prediction of STBIM.

### B.6.2 ROOT CAUSE ANALYSIS

We further visualize the node embedding and temporal embedding of STID, as shown in Figure 5. Regarding node embeddings, STID focuses on capturing shared patterns among nodes where nodes with similar traffic distributions cluster together. Hence, the node embeddings exhibit cluster distribution (Shao et al., 2022a). Clearly, these shared patterns among nodes have limited utility in distinguishing spatial inconsistency features among nodes. Concerning time embeddings, STID captures periodic features which exhibit repetitive cycles (Shao et al., 2022a), while temporal inconsistency features where traffic suddenly increases or decreases are rare. Therefore, this embedding evidently fails to capture temporal inconsistencies.

We further demonstrate some prediction cases of STBIM in Figure 6. the errors between the base prediction of STID and the labels (i.e., $\mathbf{y}_{base} - \hat{\mathbf{y}}$), and the prediction correction terms ($\mathbf{y}_{corr}$)

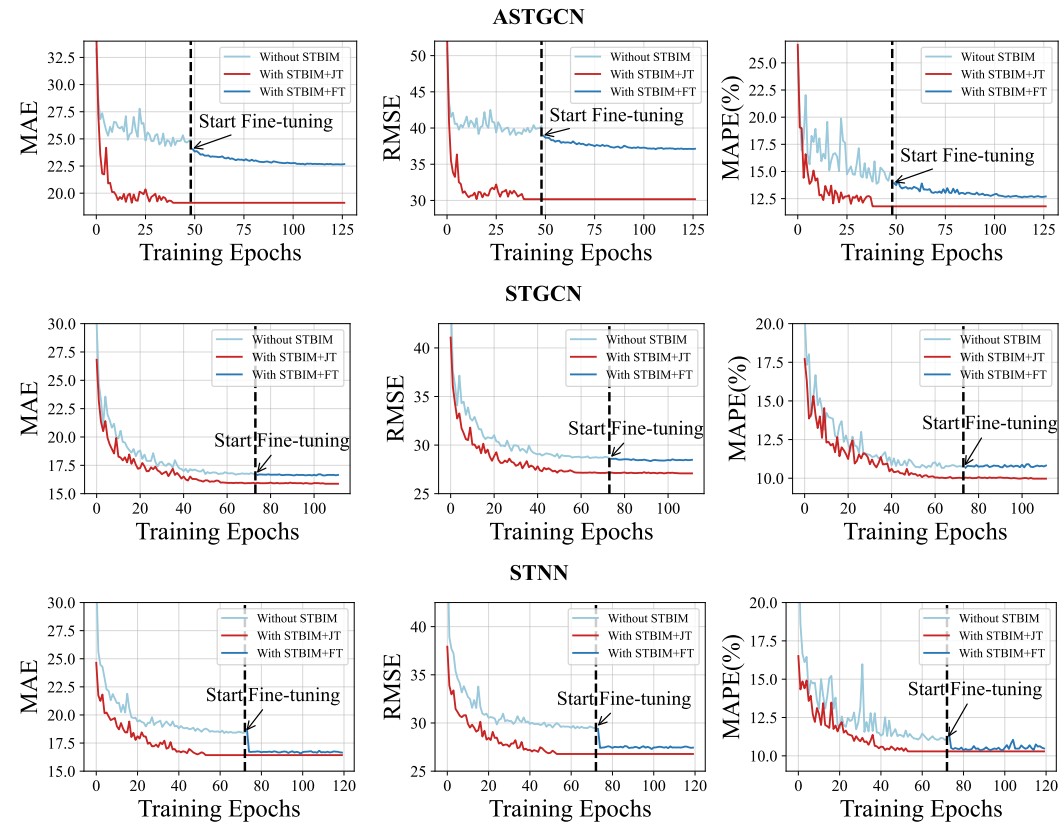

Figure 7: A comparison of the convergence speed and convergence results in validation phrase of baselines without STBIM, with STBIM+JT and with STBIM+FT on the LargeST-SD dataset.

generated by combining STID in two training modes with STBIM are shown. We can find that the correction terms produced by STBIM can fit the bias of the base prediction, illustrating the improved handling of spatiotemporal inconsistencies by STBIM.

## B.7 COMPUTATIONAL COMPLEXITY

### B.7.1 EFFICIENCY ANALYSIS

In our evaluation on the LargeST-SD and LargeST-CA dataset, we present the efficiency costs associated with integrating STBIM with various advanced models. One key advantage of STBIM is observed in accelerating the convergence speed of the models. Through visualizations, we showcase the training processes of several STNNs with STBIM. By visualizing the model training procedures using the SD and Knowair-1 datasets with multiple spatiotemporal prediction as examples, as depicted in Figures 7, we observe that the integration of STBIM leads to faster convergence, especially when employing the joint training strategy. This acceleration in convergence speed can be attributed to the improved model fitting to the data distribution by correcting prediction, thereby alleviating the learning burden caused by redundant features.

Furthermore, we delve into the computational complexity of STBIM by examining several advanced space-time prediction models, as illustrated in Table 13. Our analysis reveals that STBIM offers substantial performance gains at a relatively modest model parameter cost. To optimize training efficiency, we utilize the fine-tuning training method by directly fine-tuning the pre-trained STNN with STBIM, thereby minimizing time overhead. In conclusion, given the significant performance enhancements, the complexity burden introduced by STBIM is considered acceptable. It is important to note that our primary focus lies on the model's performance rather than efficiency, which differs from the current emphasis in the spatiotemporal community.

Table 13: Model efficiency analysis on the LargeST-SD dataset. We report the improvement in average prediction performance over MAPE.

| Method | | Parameter | Train time/epoch(s) | Total train time (h) | Inference time (s) | Memroy (MB) | Improvement |
|---|---|---|---|---|---|---|---|
| Baseline | STBIM | | | | | | |
| STGCN | - | 508K | 90.89 | 2.09 | 12.83 | 3523 | - |
| | + JT | 624K | 148.64 | 3.52 | 20.47 | 6180 | +8.79% |
| | + FT | 624K | 108 | 1.86 | 20.47 | 4202 | +8.13% |
| STID | - | 128K | 7.69 | 0.32 | 1.61 | 980 | - |
| | + JT | 244K | 12.27 | 0.39 | 2.16 | 1520 | +6.06% |
| | + FT | 244K | 8.50 | 0.18 | 2.16 | 1176 | +1.24% |
| ASTGCN | - | 2.2M | 466.95 | 7.36 | 28.47 | 10396 | - |
| | + JT | 2.6M | 900.16 | 10.23 | 50.39 | 22878 | + 14.13% |
| | + FT | 2.6M | 505.66 | 12.09 | 50.39 | 15727 | +8.68% |
| AGCRN | - | 761K | 365.29 | 7.63 | 32.45 | 7827 | - |
| | + JT | 1.0M | 609.40 | 11.92 | 55.29 | 13827 | +9.57% |
| | + FT | 1.0M | 455.57 | 1.97 | 55.29 | 8080 | + 0.08% |

Table 14: Model efficiency analysis on the LargeST-CA dataset. We report the improvement in average prediction performance over MAPE.

| Method | | Parameter | Train time/epoch(s) | Total train time (h) | Inference time (s) | Memroy (MB) | Improvement |
|---|---|---|---|---|---|---|---|
| Baseline | STBIM | | | | | | |
| STGCN | - | 508K | 788.75 | 30.21 | 236.77 | 29470 | - |
| | +JT | 624K | 1319.47 | 55.19 | 271.07 | 53156 | +7.60% |
| | +FT | 624K | 882.39 | 33.52 | 271.07 | 31615 | +5.63% |
| STID | - | 150K | 232.95 | 4.94 | 55.15 | 6704 | |
| | +JT | 270K | 303.17 | 9.23 | 72.38 | 11265 | +12.80% |
| | +FT | 270K | 276.38 | 5.58 | 72.38 | 7261 | +14.61% |

### B.7.2 Further Analysis

Using STID and $D^2$STGNN datasets as example, we aim to demonstrate that the performance improvement brought by STBIM does not stem from increasing parameter size or computational efforts, which can not achieve the same effect of our STBIM, we have create a variant which stacks two STNNs (STID and $D^2$STGNN) to increase the parameter size, and we increase computational effort by training STNNs using double maximum epochs and the patience of the early stop strategy, and this variant is defined as STNN-Plus. The experimental results are shown in Table 15, and it is evident that simply increasing complexity does not provide substantial benefits compared to STBIM. For certain complex models, such as $D^2$STGNN, an excessive parameter size can result in overfitting. This observation highlights that while more parameterized models may have the potential to extract richer features from input data, they still fail to adequately approximate label features. Consequently, the input-label consistency assumption inherent in existing spatiotemporal architectures is inherently fragile, underscoring the critical need to explicitly model label features in order to enhance predictive performance.

Table 15: Average performance comparison of STNN, STNN-Plus, and STNN+STBIM on LargeST-SD dataset.

| Model | MAE | RMSE | MAPE |
|---|---|---|---|
| STID | 18.00 | 30.75 | 12.05 |
| STID-Plus | 17.94 | 30.43 | 12.13 |
| STID+STBIM | 17.08 | 28.92 | 11.32 |
| $D^2$STGNN | 17.44 | 29.58 | 12.18 |
| $D^2$STGNN-Plus | 17.68 | 31.04 | 12.57 |
| $D^2$STGNN+STBIM | 17.19 | 28.53 | 11.20 |

## C  RELATED WORK OF SPATIOTEMPORAL SHIFT

Traditional spatiotemporal architectures adhere to the independent and identically distributed (IID) assumption, while spatiotemporal data shift poses a challenge for out-of-distribution generalization; however, spatiotemporal data distribution shift poses a challenge for out-of-distribution (OOD) generalization. Several spatiotemporal out-of-distribution (OOD) models have emerged in recent literature. For instance, CauSTG (Zhou et al., 2023) introduces a causal framework designed to transfer both local and global spatiotemporal invariant relationships to OOD scenarios. CaST (Xia et al., 2023) utilizes a structural causal model (SCM) to interpret the data generation processes of spatiotemporal graphs, employing back-door adjustment techniques to isolate invariant components from the temporal environment. Similarly, STEVE (Hu et al., 2023) encodes traffic data into two disentangled representations and incorporates spatiotemporal environments as self-supervised signals, thereby integrating contextual information into these representations. Additionally, STONE (Wang et al., 2024) proposes a causal graph structure aimed at learning robust spatiotemporal semantic graphs for OOD learning. However, while these models focus on addressing overall shifts between training and testing data, we focus on a more granular shift between historical observed data (input) and predicted data. This shift is present in both OOD and IID scenarios.

### C.0.1  ABLATION EXPERIMENT

We conduct ablation experiments on the Large-SD dataset using the STGCN. We created two variants: "w/o MLP" means we remove the retrospect MLP, and "w/o kernel" means that we remove the propagation kernel for smoothing. The experimental results are shown in Table 16. "w/o MLP" showed significantly higher errors, as this retrospect MLP is used to map label features to the same hidden space as input features, leading to smoother training. The experiment without the kernel resulted in poor prediction performance because residual smoothing benefits the model's learning process.

Table 16: Ablation experiment.

| STGCN | | | |
| --- | --- | --- | --- |
| Model | MAE | RMSE | MAPE |
| w/o MLP | 19.24 | 32.63 | 12.99 |
| w/o kernel | 18.75 | 32.38 | 12.62 |
| STBIM | **18.41** | **31.84** | **12.45** |
| D2STGNN | | | |
| Model | MAE | RMSE | MAPE |
| w/o MLP | 19.57 | 33.44 | 13.80 |
| w/o kernel | 18.95 | 32.65 | 12.68 |
| STBIM | **17.19** | **28.53** | **11.20** |

## D  DISCUSSION

we develop a generic module that can be combined with STNNs to improve their learning ability. In this section, we discuss the limitations of the work, which will serve as future research directions.

Firstly, we will explore more expressive models for label correlation modeling. In this paper, we used simple GCNs to capture spatiotemporal correlations in labels, which were then used to correct the model's prediction. In the future, we aim to explore more complex models to further enhance performance.

Second, we discussed four most commonly used kernel functions for residual information propagation in this paper. In fact, in spatiotemporal graph learning, there are also some other methods. For example, some works (Li and Zhu, 2021; Lan et al., 2022) represent the kernel as the similarity of data distributions between nodes. In the future, we will explore a broader range of kernel methods.

