# OpenReview forum: "Spatiotemporal Backward Inconsistency Learning Gives STGNNs Icing on the Cake"
_ICLR.cc/2025/Conference — ICLR 2025 Conference Withdrawn Submission_

### Official Review · Reviewer_tHSS · 2024-11-03

**Soundness:** 2
**Presentation:** 2
**Contribution:** 2
**Rating:** 3
**Confidence:** 4

**Summary:**

The paper proposes Spatio-Temporal Backward Inconsistency Learning Module (STBIM) designed to enhance Spatiotemporal Neural Networks (STNNs) by addressing the problem of input-label inconsistency in spatiotemporal prediction tasks. STBIM operates by capturing residuals between input data and labels through their spatio-temporal features, smoothing these residuals through a residual propagation kernel, and adding the resulting corrections to the base predictions generated by STNNs. Extensive experiments across multiple datasets and baseline models demonstrate substantial improvements in prediction accuracy.

**Strengths:**

* The experimental validation across 11 datasets with diverse STNN architectures showcases the generalizability and effectiveness of STBIM. Performance gains up to 21% underscore its practical impact.
* The fact that STBIM is designed to be compatible with a wide range of STNN architectures enhances its potential for broader application and scalability in different domains like traffic and atmospheric forecasting.

**Weaknesses:**

* The method for measuring the inconsistency between the input data and labels seems redundant in spatio-temporal methods. Ideally, the spatio-temporal methods such as convolution or transformer and their temporal variants (convLSTM or spatio-temporal transformer) have the inductive biases required to take into account the spatio-temporal features that are mapped to labels. It is just a matter of adding additional layers that anyway this method proposes to use (STBIM is a separate module). I would suggest the authors to provide a more detailed comparison between STBIM and traditional spatiotemporal architectures, specifically addressing how STBIM's approach differs from or improves upon simply adding more layers. Further providing clarification on how STBIM's method of measuring inconsistency provides information that cannot be captured by the inductive biases of existing spatiotemporal methods would help the paper.
* Since, there is no new information added (such as test-time labels), adding just an inconsistency loss should not bring any new information for such impressive performance gains and those gains should be achievable through traditional architectures as well. Conducting an ablation study comparing STBIM to equivalent increases in model complexity for traditional architectures. This would help demonstrate why traditional architectures cannot achieve similar gains. I would suggest the authors explain in more detail how STBIM extracts additional information from the existing data that traditional architectures may miss.
* Although performance gains are highlighted, the computational cost associated with adding STBIM to complex STNN models is not fully explored. An analysis of training and inference times with and without STBIM would clarify its practical feasibility. A detailed computational cost analysis, including training and inference times, for models with and without STBIM across different dataset sizes and model complexities could be provided. Additionally, a breakdown of memory requirements for models with and without STBIM.

**Questions:**

* Could the authors provide an in-depth breakdown of the computational resources required to train models with and without STBIM? How does this impact its scalability?
* how much of the above performance can be recovered by using additional spatio-temporal layers?
* How will the test-time labels be used if available?
* What extra information is the inconsistency loss bringing/recovering from the input data apart from just maintaining a spatial smoothness, for which convolution / transformer have the right inductive bias?

---

> ### Author Response · Authors · 2024-11-21
> **Clarification of Weakness 1.**
>
> Dear Reviewer tHSS,
>
> Thank you very much for your valuable comments; we will respond to your concerns point by point.
>
> > (1) Simply expanding the parameters cannot achieve the same gains as STBIM.
>
> **We have completely refuted this point in the original manuscript, involving the line 451 in Experiment Section 5.1**. Subsequently, we have demonstrated this point in **Appendix B.7.2** in detail. Taking the spatiotemporal backbones STID and D2STGNN as examples, we introduced variants: STID-Plus and D2STGNN-Plus, which stacked two backbones to expand the parameter scale, while also increasing the maximum training epochs. The experimental results are presented in Table 14 of the manuscript, which we have replicated below for convenience.
>
> |               | MAE    | RMSE  | MAPE  |
> |:-------------:|:------:|:-----:|:-----:|
> | STID          | 18.00  | 30.75 | 12.05 |
> | STID-PLUS     | 17.94  | 30.43 | 12.13 |
> | STID+ours     | 17.08  | 29.92 | 11.32 |
> |               |        |       |       |
> | D2STGNN       | 17.44  | 29.58 | 12.18 |
> | D2STGNN-Plus  | 17.68  | 31.04 | 12.57 |
> | D2STGNN+STBIM | 17.19  | 28.53 | 11.20 |
>
> We can observe that simply increasing the parameter size does not lead to performance enhancement, For the complex model D2STGNN, simply stacking models may actually decrease model performance, as an excessively large parameter size can lead to overftting of the model to the data. Our performance improvement originates from the effective modeling of inconsistent features.
>
>  Please refer to the experimental analysis section of the main body and Appendix B.7.1 in the submitted manuscript, we have discussed this when we submitted the paper.
>
> ----
>
>
> >  "The method for measuring the inconsistency between the input data and labels seems redundant?"
>
> **We argue that the method of modeling the inconsistency between input data and labels is not redundant !** On the contrary, it is a crucial factor contributing to our performance. **In Experiment Section 5.3 and Appendix Section B.6 of the manuscript**, we evaluated the performance of various spatiotemporal models on spatiotemporal inconsistency samples. Clearly, STBIM can effectively enhance these models to better handle such inconsistencies. For example, as shown in Table 4, taking STID as an example, STBIM can improve its performance on handling inconsistent samples by 24.05%, significantly boosting the predictive performance of STID.

---

> ### Author Response · Authors · 2024-11-21
> **Clarification of Weakness 2.**
>
> > (1) ' Adding just an inconsistency loss should not bring any new information for such impressive performance gains'
>
> In our method, we did not add any additional losses except the regression loss.
>
> > (2).  ‘Those gains should be achievable through traditional architectures as well’
>
> Unfortunately, even if an STGNN is designed with a more sophisticated structure, they do not emphasize modeling the inconsistency between input and labels. STBIM, on the other hand, addresses this issue and achieves performance improvements by focusing on it. Through extensive experimental cases (approximately 15 spatiotemporal prediction models), we demonstrate that STGNNs can benefit from STBIM to gain additional potential performance improvements, with performance increases of up to 18.74%, even for advanced models. As mentioned earlier, this gain cannot be achieved through simply expanding the parameter scale.
>
> > (3) ' Conducting an ablation study comparing STBIM to equivalent increases in model complexity for traditional architectures. ’
>
> Increasing model parameters and computational complexity is not equivalent to the benefits brought by STBIM, as discussed above.
>
> > (4) What information traditional architectures may miss or No additional information?
>
> This is where our design gets magical.
>
> These traditional architectures cannot effectively model label features, leading to confused predictions for samples with inconsistent input labels, as discussed in our introduction. **This is where our clever design comes in - we noticed this key information about label representation, which contains abundant table features, enabling the model to explicitly utilize label features during training to enhance learning from inconsistent samples without additional information, thereby improving the accuracy of the inference process.** We validated this motivation through extensive experiments in the Experiment Section 5.3 and Appendix Section B.6 of the manuscript.

---

> ### Author Response · Authors · 2024-11-21
> **Clarification of Weakness 3: Computational complexity.**
>
> **We have reported the computational overhead introduced by STBIM in Appendix B.7 (as indicated in the main manuscript at line 347)**. In the paper, we present the parameter scale, training time, and performance improvements achieved. To address your concerns further, we additionally report the computational complexity on CA dataset with 8600 nodes to demonstrate the scalability of STBIM on the large-scale scenario. For your convenience, we have replicated the results for the CA dataset in the table below as an example:
>
> | STGCN | Parameters  | Train time/epoch(s) | Total train time (h) | Inference time (s) | Memroy (MB) |
> |:-----:|:-----------:|:-------------------:|:--------------------:|:------------------:|:-----------:|
> | -     | 508K        | 781                 | 30.21                |                    | 29470       |
> | +JT   | 624K        | 1321                | 55.19                |                    | 53156       |
> | +FT   | 624K        | 882                 | 33.52                |                    | 31615       |
> |       |             |                     |                      |                    |             |
> | STID  | Parameters  | Train time/epoch(s) | Total train time (h) | Inference time (s) | Memory (MB) |
> | -     | 150K        | 232                 | 4.94                 |                    | 6204        |
> | +JT   | 270K        | 303                 | 9.23                 |                    | 11865       |
> | +FT   | 270K        | 276                 | 5.58                 |                    | 7261        |
>
> Our analysis shows that STBIM achieves significant performance gains with minimal computational complexity. Fine-tuning training methods involve less computational time by directly fine-tuning pre-trained STNN with STBIM. Both approaches balance performance and efficiency, offering flexibility in selection. In conclusion, given the notable performance enhancement, the complexity burden introduced by STBIM is considered acceptable. Please note that we have analyzed that traditional architectures cannot simply directly increase the computational complexity to achieve similar performance gains.

---

> ### Author Response · Authors · 2024-11-21
> **Clarification of question**
>
> > Q1. Computational complexity.
>
> Please refer to Weakness 3.
>
> > Q2. how much of the above performance can be recovered by using additional spatio-temporal layers?
>
> Simply expanding the parameters cannot achieve the same gains as STBIM.
>
> **We have completely refuted this point in the original manuscript, involving the line 451 in Experiment Section 5.1**. Subsequently, we have demonstrated this point in **Appendix B.7.2** in detail. Taking the spatiotemporal backbones STID and D2STGNN as examples, we introduced variants: STID-Plus and D2STGNN-Plus, which stacked two backbones to expand the parameter scale, while also increasing the maximum training epochs. The experimental results are presented in Table 14 of the manuscript, which we have replicated below for convenience.
>
> |               | MAE    | RMSE  | MAPE  |
> |:-------------:|:------:|:-----:|:-----:|
> | STID          | 18.00  | 30.75 | 12.05 |
> | STID-PLUS     | 17.94  | 30.43 | 12.13 |
> | STID+ours     | 17.08  | 29.92 | 11.32 |
> |               |        |       |       |
> | D2STGNN       | 17.44  | 29.58 | 12.18 |
> | D2STGNN-Plus  | 17.68  | 31.04 | 12.57 |
> | D2STGNN+STBIM | 17.19  | 28.53 | 11.20 |
>
> We can observe that simply increasing the parameter size does not lead to performance enhancement, For the complex model D2STGNN, simply stacking models may actually decrease model performance, as an excessively large parameter size can lead to overftting of the model to the data. Our performance improvement originates from the effective modeling of inconsistent features.
>
> > Q3.  How will the test-time labels be used if available?
>
> According to the general paradigm of machine learning, labels are not accessible during testing. While labels are available during training, they cannot be directly used as inputs to the model. Typically, we optimize the model by utilizing backpropagation with label information loss.
>
> > Q4. What extra information is the inconsistency loss bringing/recovering from the input data apart from just maintaining a spatial smoothness？
>
> Our inconsistency modeling (not a loss function) can incorporate label features into the model without using additional inputs, improving the model's prediction performance on inconsistent samples. In addition to maintaining spatial smoothness, it also preserves temporal smoothness.
>
> **We evaluated the effectiveness of STBIM in helping models to eliminate inconsistency between input and label dimensions in the temporal dimension in Section 5.3 and Appendix B.5**, thereby ensuring temporal smoothness of the backbone, and the results prove that our proposed module STBIM comprehensively deals with the inconsistencies of temporal and spatial dimensions.

---

### Official Review · Reviewer_FGB6 · 2024-11-04

**Soundness:** 2
**Presentation:** 2
**Contribution:** 3
**Rating:** 5
**Confidence:** 3

**Summary:**

The paper introduces a Spatio-Temporal Backward Inconsistency Learning Module (STBIM) designed to enhance spatiotemporal prediction models by addressing input-label inconsistencies. This approach incorporates a residual learning mechanism to refine predictions and improve performance across multiple domains, such as traffic and climate prediction. STBIM’s integration demonstrates significant accuracy improvements in various datasets and model types.

**Strengths:**

1. The paper presents a novel paradigm for handling "input-label inconsistencies" with residual theory based on spatiotemporal Gaussian Markov Random Field.
2. The proposed STBIM is model-agnostic, allowing easy integration into existing models without extensive modification.
3. Experiments demonstrate versatility through extensive testing across diverse datasets and models.

**Weaknesses:**

1. The paper lacks a thorough discussion of related works addressing spatiotemporal inconsistency. While the discussion on "Label propagation in GNN," does not directly address the core issue and could be streamlined.
2. The connection between the spatiotemporal residual theory and the proposed STBIM module is somewhat unclear. See questions for details.
3. The paper does not include an ablation study for the individual components of STBIM, such as the retrospect MLP and the propagation kernel for smoothing.

**Questions:**

1. The paper introduces the unique concept of "input-label inconsistencies" in spatiotemporal prediction. Could you clarify how this concept differs from other types of spatiotemporal inconsistency, such as spatiotemporal out-of-distribution (OOD) or distribution shift?
2. The current claim regarding differences between input and label features (Line 269) seems ambiguous. While the paper uses hidden embeddings as features for input and label, the original values can also be considered features. Does this imply that the difference between input and label values could serve as the residual (if labels are available)? This seems contradictory to Equation 8, which defines the residual as the difference between prediction and label. Could you clarify?
3. In the case of “similar input data resulting in different labels," if there are two identical inputs with different labels, wouldn’t the method struggle to predict correctly since it is a deterministic model? Can this limitation be addressed?
4. What is STRIP in Figure 4(b)?
5. Can the authors provide insights into the computational overhead introduced by STBIM, particularly in large-scale applications?

---

> ### Author Response · Authors · 2024-11-21
> **Clarification of weakness**
>
> Dear Reviewer FGB6,
>
> Thank you very much for your valuable review; it is crucial for improving the quality of our manuscript.
>
> **W1**. Discussion of related works addressing spatiotemporal inconsistency
>
> The inconsistency we define differs fundamentally from the concept of shift in spatiotemporal out-of-distribution (OOD) learning.  The latter refers to the overall differences in data distribution between training and testing datasets, which can lead to OOD challenges.  In contrast, our focus is on a more granular aspect: the inconsistency between input values and future values.  This inconsistency can occur not only in OOD scenarios but also in independent and identically distributed (IID) situations.
>
> To further assess the impact of the proposed Spatiotemporal Bidirectional Interpolation Model (STBIM), we investigate whether spatiotemporal shift learning models can benefit from its implementation. Following the OOD setting in the open-source code of STONE [1], we use their STSD dataset as example and select two spatiotemporal OOD learning models as backbones, and STBIM is integrated with joint-training manner. The results are as follows:
>
> |                           |           |        |          |
> |:------:|:-----:|:-----:|:-----:|
> |        | MAE   | RMSE  | MAPE  |
> | STONE  | 18.46 | 30.65 | 15.29 |
> | +STBIM | **17.89** | **30.08** | **14.91** |
> | CaST   | 26.77 | 40.20 | 21.48 |
> | +STBIM | **25.63** | **37.86** | **20.95** |
> |                           |           |        |          |
>
> We find that STBIM can also improve the predictive performance of spatiotemporal shift learning models because it effectively enhances the model's accuracy on samples with historical-future inconsistency in OOD scenario. Following your suggestion, we have deleted the introduction about "Label propagation in GNN” and included this discussion in Section Related Work in the revised version.
>
> ---
> **W2**. The details of spatiotemporal residual theory.
>
> Yes, if the labels are available, the difference between the input and the label is used as residual input to the model for learning. However, this assumption is never true because, according to the general paradigm of machine learning, labels cannot be directly used as model inputs for regression prediction tasks. Therefore, we propose an effective solution using label hidden embeddings, and these high-dimensional representations contain richer feature information to comprehensively learn the residual information.
>
> Equation 8 in the paper defines the residual as the difference between the predicted expectation and the label expectation. To analyze the internal patterns of spatiotemporal data, we employ the Gaussian Markov Random Field (GMRF) model. In this framework, **as discussed in lines 268 to 276 of our manuscript, the prediction expectation is determined by the high-dimensional representation of the input**, which arises from the spatiotemporal learning model's abstraction of input features, as illustrated in Equation 6. Conversely, the **label expectation is governed by its intrinsic characteristics**. Thus, at a fundamental level, the difference between these two expectations approximates the disparity between input features and label features. We will polish this discussion for clarity.
>
> ---
>
> **W3**.Ablation study
>
> Sorry for the confusion. We conduct ablation experiments on the Large-SD dataset using the STGCN. We created two variants: "w/o MLP" means we remove the retrospect MLP, and "w/o kernel" means that we remove the propagation kernel for smoothing. The experimental results are as follows:
> | Large-SD   | MAE       | RMSE   | MAPE  |
> |------------|-----------|--------|-------|
> | w/o MLP    | 19.24     | 32.63  | 12.99 |
> | w/o kernel | 18.95     | 32.65  | 12.68 |
> | +STBIM     | 18 41 | 31.84  | 12.45 |
>
> "w/o MLP" showed significantly higher errors, as this retrospect MLP is used to map label features to the same hidden space as input features, leading to smoother training. The experiment without the kernel resulted in poor prediction performance because residual smoothing benefits the model's learning process. We would include the ablation experiments with more backbones in the revised version.

---

> ### Author Response · Authors · 2024-11-21
> **Clarification of question**
>
> **Q1.** The paper introduces the unique concept of "input-label inconsistencies" in spatiotemporal prediction. Could you clarify how this concept differs from other types of spatiotemporal inconsistency, such as spatiotemporal out-of-distribution (OOD) or distribution shift? **Please refer to W1.**
>
> **Q2.** The current claim regarding differences between input and label features (Line 269) seems ambiguous. **Please refer to W2.**
>
>
> **Q3.**  Similar input data resulting in different labels.
>
> As emphasized in the introduction, this dilemma is a critical challenge faced by traditional spatiotemporal learning models.    In cases where two nodes exhibit similar input distributions but have different labels, the model struggles to effectively differentiate between these nodes, thus resulting in similar predictions for both and leading to substantial errors between predictions and labels.    This observation serves as our motivation: to explicitly model label features in order to enhance the ability of spatiotemporal learning models to address samples with similar inputs but differing labels, as well as samples with different inputs but similar labels.    We will revise the corresponding content in the introduction to further emphasize this motivation.
>
> **Q4.** What is STRIP in Figure 4(b)?
>
> Sorry for the error, it should be corrected to STBIM here. And we have corrected this in the revised version.
>
> **Q5.**  Computational stduy
>
> **We have reported the computational overhead introduced by STBIM in Appendix B.7 (as indicated in the main manuscript at line 347) when we submitted the paper.** To address your further concerns, we additionally report the computational complexity on LargeST-CA dataset with 8600 nodes to demonstrate the scalability of STBIM on the large-scale scenario. For your convenience, we have replicated the results for the LargeST-CA dataset in the table below as an example:
>
> | STGCN | Parameters  | Train time/epoch(s) | Total train time (h) | Inference time (s) | Memroy (MB) |Improvement
> |:-----:|:-----------:|:-------------------:|:--------------------:|:------------------:|:-----------:|:-----------:|
> | -     | 508K        | 788.75                 | 30.21                |         236.72           | 29470       |-
> | +JT   | 624K        | 1319.47                | 55.19                |           271.07         | 53156       |+7.60%
> | +FT   | 624K        | 882.39                 | 33.52                |        254.41            | 31615       |+5.63%
> |       |             |                     |                      |                    |             |
> | STID  | Parameters  | Train time/epoch(s) | Total train time (h) | Inference time (s) | Memory (MB) |
> | -     | 150K        | 232.95                 | 4.94                 |          55.15          | 6704        |-
> | +JT   | 270K        | 303.17                 | 9.23                 |         72.38           | 11265       |+12.80%
> | +FT   | 270K        | 276.38                 | 5.58                 |         70.24           | 7261        |+14.61%
>
> Our analysis shows that STBIM achieves significant performance gains with minimal computational complexity. Fine-tuning training methods involve less computational time by directly fine-tuning pre-trained STNN with STBIM. Both approaches balance performance and efficiency, offering flexibility in selection. In conclusion, given the notable performance enhancement, the complexity burden introduced by STBIM is considered acceptable. Please note that in **Appendix B.7.2**, we also analyzed that traditional architectures cannot simply directly increase the computational complexity to achieve similar performance gains.

---

> > ### Comment · Reviewer_FGB6 · 2024-11-21
> >
> > W1: I may be misunderstanding something, but your claim about OOD is confusing. You state: "The latter refers to the overall differences in data distribution between training and testing datasets, which can lead to OOD challenges. In contrast, our focus is on a more granular aspect: the inconsistency between input values and future values." However, a typical case of OOD involves training on historical data and testing on future data, where distribution changes occur—temporal OOD/shift. Could you clarify how your focus differs from this understanding?
> >
> > W2: Of course we cannot use ground truth labels for prediction. However, your high-dimensional representations are not embeddings of the ground truth labels but rather embeddings of the predicted labels. If that's correct, this "residual information" essentially reflects the difference between input values and predicted labels. If so, I struggle to understand how this provides meaningful information. Could you elaborate?
> >
> > Q3: Could you answer my question directly? Can your method make correct predictions if two identical inputs are given but have different labels?

---

> > > ### Author Response · Authors · 2024-11-22
> > > **Thank you very much for your response.**
> > >
> > > > W1. Shift in OOD.
> > >
> > > When executing our prediction model, both test and training sets are divided along the temporal axis into a set of input values and labels. For example, in the test set, we select data from $T_p$ past time steps at time point t, $X_t \in R^{T_{p}\times N}=[x_{t-T_p+1},…, x_t]$ as input values, and then select data from the following $T_f$ future time steps $Y_t  \in R^{T_{p}\times N}=[x_{t-T_f+1},…, x_t]$ as prediction values. Through this operation, the test set essentially becomes a union of a set of input values $\mathbb{X}=[X_0,...,X_L]$ and a set of labels $\mathbb{Y}=[Y_0,...,Y_L]$, where $L$ means the number of samples. Test set is denoted as $\mathbb{D}$=$\mathbb{X}$$\bigcup$$\mathbb{Y}$. After similar operations, the training is recorded as $\mathbb{T}$.
> > >
> > > **We focus on the differences between inputs and labels**, but inputs and labels are sampled from the same distribution. This difference reflected in two aspects:（1）The difference  in the node dimension. Given the input of node $v$ and node $u$ at time step t: $X_t^u$ and $X_t^v$ where $X_t^u$ and $X_t^v$ is similar, but the label of two nodes is significantly different. （2）The difference  in the temporal dimension. Given the input and label of node v$ at time step t : $X_t^u$ and $Y_t^u$, their statistical characteristics (such as mean and variance) differ significantly.
> > >
> > > **OOD tasks focus on the differences between test set distribution** Pr($\mathbb{D}$) and training set Pr($\mathbb{T}$), i.e., Pr($\mathbb{D}$) $\neq$ Pr($\mathbb{T}$). which we refer to as overall differences. The differences we are concerned about also exist in the OOD task.
> > >
> > > > W2. Label embedding
> > >
> > > Yes, during the training process, as the model's predictions become increasingly accurate, the embeddings of model-generated predictions also approach the mapping of true values (though gaps still exist). At this point, the prediction embeddings also contain numerous label features, and we make the most use of these features to improve model accuracy.
> > >
> > > > W3. Could you answer my question directly? Can your method make correct predictions if two identical inputs are given but have different labels?
> > >
> > > Yes, we can do better, as shown in Figure 4. Allow me to further clarify your misunderstanding. The inconsistency we focus on refers to cases within a sample (containing data from T time steps for N nodes) where (some) nodes have the same inputs but different labels. In this case, the model will tend to make similar predictions for these nodes, which can lead to errors.
> > >
> > > If two samples that have identical values, meaning all N×T data points are exactly the same, all models will give identical predictions - I believe this is what you refer to as deterministic models.

---

> > > > ### Comment · Reviewer_FGB6 · 2024-11-24
> > > >
> > > > I still find the definition of input-label differences/residuals to be problematic. As I mentioned earlier, computing differences between input values and predicted labels (or even true labels) makes no sense to me.
> > > >
> > > > Regarding W3, consider two spatiotemporal graphs (samples) that are nearly identical except for one node in each graph having different future trajectories (as per your claim about the node-level model). In such a scenario, it is evident that the model would fail to make the correct prediction.

---

### Official Review · Reviewer_RjnU · 2024-11-07

**Soundness:** 3
**Presentation:** 2
**Contribution:** 3
**Rating:** 3
**Confidence:** 4

**Summary:**

This paper proposes a spatiotemporal prediction module, the spatiotemporal backward inconsistency learning module (STBIM), to solve the problem of inconsistent input labels in existing spatiotemporal neural networks (STNNs) (i.e., the same input may lead to different outputs, or different inputs may have the same output). By combining label features and integrating spatiotemporal residual theory, STBIM effectively improves the prediction accuracy of the model. Experimental results show that STBIM significantly improves the prediction performance across multiple datasets.

**Strengths:**

1. The authors introduces an innovative module to capture spatiotemporal inconsistencies between inputs and outputs (labels in this paper).

2. The authors conduct many experiments to show the effectiveness of the proposed model STBIM.

3. The code is given which is helpful to reproduce the experimental results.

**Weaknesses:**

1. I have significant reservations about the authenticity of the experimental results. For example, on page eight, the findings for the "LargeST-GBA dataset", particularly regarding the BigST model, show that the average improvement is considerably less than the claimed increase of over seventeen percentage points.

2. The model's core concept involves correcting predicted outcomes through spatiotemporal residuals. This notion closely resembles shift learning in spatiotemporal networks, yet there is no discussion of the shifts.

3. Additionally, the writing quality in the article requires improvement, as evidenced by a typographical error in the first line of page five ("to be be") and inconsistencies in tense and capitalization throughout.

**Questions:**

1. The third subplot in Figure 1 is not mentioned in the text, and the "abnormal signal" it shows is not referenced in the document. Is it necessary to keep it? What is its significance?

2. The text highlights the inconsistency between inputs and outputs, using the prediction label to represent the time series of a later time window. Can this be expressed more clearly? If a label is used, could the specific cases be clarified? Typically, a label is a distinct value, not a continuous range of floating-point numbers.

3. If all results are accurate, why do many fine-tuned results exceed those of joint tuning? Can this be analyzed further?

---

> ### Author Response · Authors · 2024-11-21
> **Rebuttal of Weakness**
>
> Dear Reviewer RjnU,
>
> Thank you very much for your valuable review; it is crucial for improving the quality of our manuscript.
>
> - ### W1  Reproducibility
>
> We sincerely apologize for the oversight in reporting the performance of BigST, and the actual average Mean Absolute Error (MAE) performance of BigST, which was mistakenly recorded as 24.12, is accurately represented as 21.12. **This error have occurred due to the proximity of the digits "4" and "1" on the numeric keypad of the Chinese (US) standard keyboard layout**😭, which is illustrated below.
>
> |   | Keyboard Layout |   |
> |:---:|:---------------:|:---:|
> | 7 | 8               | 9 |
> | **4** | 5               | 6 |
> |**1** | 2               | 3 |
> |  |           |        |          |
>
> Below, we present the average performance metrics for BigST over 12 time steps. The table above contains the incorrectly reported performance values, while the table below provides the corrected figures.
>
> |                           |           |        |          |
> |:-------------------------:|:---------:|:------:|:--------:|
> |                           | MAE       | RMSE   | MAPE     |
> | BigST( Incorrect results) | **24.42** | 34.54  | 18.19    |
> | BigST( Correct results)   | **21.42** | 34.54  | 18.19    |
> | +STBIM(JT)                | 20.18     | 33.34  | 15.37    |
> |         Gap                  | +5.79%    | +3.48% | +15.50%  |
> | +STBIM(FT)                | 20.16     | 33.03  | 15.45    |
> |           Gap                 | +5.88%    | +4.39% |  +15.01% |
> |                           |           |        |          |
>
> The corrected results continue to illustrate the effectiveness of STBIM, as evidenced by a 15.50% improvement in the average MAPE performance of BigST.  To address any further concerns, we will provide access to all training logs for the models via an anonymous code link for your review.  We sincerely apologize for the error caused by our oversight.
>
> ---
>
> - ### W2 Shift learning.
>
> The inconsistency we define differs fundamentally from the concept of shift in spatiotemporal out-of-distribution (OOD) learning.  The latter refers to the overall differences in data distribution between training and testing datasets, which can lead to OOD challenges.  In contrast, our focus is on a more granular aspect: the inconsistency between input values and future values.  This inconsistency can occur not only in OOD scenarios but also in independent and identically distributed (IID) situations.
>
> To further assess the impact of the proposed Spatiotemporal Bidirectional Interpolation Model (STBIM), we investigate whether spatiotemporal shift learning models can benefit from its implementation. Following the OOD setting in the open-source code of STONE [1], we use their STSD dataset as example and select two spatiotemporal OOD learning models as backbones, and STBIM is integrated with joint-training manner. The results are as follows:
>
> |                           |           |        |          |
> |:------:|:-----:|:-----:|:-----:|
> |        | MAE   | RMSE  | MAPE  |
> | STONE [1]  | 18.46 | 30.65 | 15.29 |
> | +STBIM | **17.89** | **30.08** | **14.91** |
> | CauST [2]   | 26.77 | 40.20 | 21.48 |
> | +STBIM | **24.63** | **37.86** | **20.35** |
> |                           |           |        |          |
>
> We find that STBIM can also improve the predictive performance of spatiotemporal shift learning models because it effectively enhances the model's accuracy on samples with historical-future inconsistency in OOD scenario. We w included the discussion of spatiotemporal shift learning in Section Related Work of the revised version.
>
> ---
>
> - ### W3  Writing quality.
>
> Thank you very much for your suggestion. We have enlisted the assistance of advanced language models skilled in language refinement, as well as collaborators who are native English speakers, to thoroughly proofread the manuscript and correct any grammatical errors. We appreciate your guidance in enhancing the quality of our work.

---

> ### Author Response · Authors · 2024-11-21
> **Clarification of the question**
>
> > Q1 The third subplot in Figure 1 is not mentioned in the text, and the "abnormal signal" it shows is not referenced in the document. Is it necessary to keep it? What is its significance?
>
>  We apologize for any confusion. Abnormal signals are defined as samples in which traffic flow experiences a rapid increase or decrease, highlighting a significant example of inconsistencies in the temporal dimension between input and future sequences. To elucidate this point, we would revise the original text to: "As illustrated in Figure 1 (c), inconsistencies in temporal features exist between historical and future values. Typical examples of this phenomenon include abnormal signals characterized by a rapid increase or decrease in traffic flow."
>
> ---
> > Q2 The text highlights the inconsistency between inputs and outputs, using the prediction label to represent the time series of a later time window. Can this be expressed more clearly? If a label is used, could the specific cases be clarified? Typically, a label is a distinct value, not a continuous range of floating-point numbers.
>
> Thank you very much for your suggestion. In this paper, we use 'label' to represent the time series for future time windows. To avoid further confusion, we will replace "input" and "label" with "historical value" and "future value," respectively. Consequently, the term "input-label inconsistency" will be revised to "historical-future inconsistency." To prevent other reviewers from being confused by the renaming of key concepts, we promise that this change will be consistently reflected in the revised version of the manuscript.
>
> ---
> Q3. If all results are accurate, why do many fine-tuned results exceed those of joint tuning? Can this be analyzed further?
>
> The fine-tuning method entails adjusting the STBIM and the pre-trained backbone, which serves to establish a robust optimization starting point while simplifying the optimization process.    This approach is beneficial for complex models like DGCRN and DDGCRN.    As a result, it can outperform joint training methods in specific scenarios.    Nevertheless, in the majority of cases, joint training—where STBIM and the  backbone are trained simultaneously—facilitates more flexible tuning and adaptation to the task.

---

### Note · Authors · 2024-12-03

I have read and agree with the venue's withdrawal policy on behalf of myself and my co-authors.